# MULTI-OBJECTIVE ONLINE LEARNING

## ABSTRACT

This paper presents a systematic study of multi-objective online learning. We first formulate the framework of Multi-Objective Online Convex Optimization, which encompasses a novel multi-objective dynamic regret in the unconstrained max-min form. We show that it is equivalent to the regret commonly used in the zero-order multi-objective bandit setting and overcomes the problem that the latter is hard to optimize via first-order gradient-based methods. Then we propose the Online Mirror Multiple Descent algorithm with two variants, which computes the composite gradient using either the vanilla min-norm solver or a newly designed $L_1$-regularized min-norm solver. We further derive regret bounds of both variants and show that the $L_1$-regularized variant enjoys a lower bound. Extensive experiments demonstrate the effectiveness of the proposed algorithm and verify the theoretical advantage of the $L_1$-regularized variant.

## 1 INTRODUCTION

Traditional optimization methods for machine learning are usually designed to optimize a single objective. However, in many real-world machine learning applications, we are often required to optimize multiple correlated objectives simultaneously. For example, in autonomous driving (Huang et al., 2019; Lu et al., 2019b), the self-driving vehicle needs to learn to solve multiple tasks such as self-localization and object identification at the same time. In online advertising (Ma et al., 2018a;b), the advertiser aims to choose the exposure of items to different users so that both the Click-Through Rate (CTR) and the Post-Click Conversion Rate (CVR) are maximized simultaneously. In many multi-objective scenarios, the objectives may conflict with each other (Kendall et al., 2018). Hence, there may not exist any single solution that optimizes all the objectives simultaneously. For example, in the online advertising scenario, merely optimizing either CVR or CTR will incur the degradation of performance of the other (Ma et al., 2018a;b).

Multi-objective optimization (MOO) (Marler and Arora, 2004; Deb, 2014) is concerned with optimizing multiple conflicting objectives simultaneously. Many different approaches for MOO have been proposed, which include evolutionary methods (Murata et al., 1995; Zitzler and Thiele, 1999), scalarization methods (Fliege and Svaiter, 2000) and gradient-based iterative methods (Désidéri, 2012). Recently, due to the great success of training multi-task deep neural networks, the first-order gradient-based iterative methods, i.e., Multiple Gradient Descent Algorithm (MGDA) and its variants, have regained a significant amount of research interest (Sener and Koltun, 2018; Lin et al., 2019; Yu et al., 2020). These methods compute a composite gradient based on the gradient information of all the individual objectives and then apply the composite gradient to model update. The determination of the composite gradient is based on a min-norm solver (Désidéri, 2012) which yields the common descent direction of all the objectives.

However, compared to the increasingly wide application prospect, the first-order gradient-based iterative algorithms are relatively understudied, especially for the online learning setting. Multi-objective online learning is very important due to reasons in two main aspects. First, due to the data explosion in many real-world scenarios such as various web applications, making in-time prediction requires to perform online learning. Second, the theoretical investigation will lay a solid foundation for the design of new optimizers of multi-task deep neural networks, such as multi-objective Adam.

In this paper, we conduct a systematic study of multi-objective online learning. To begin with, we first formulate the framework of Multi-Objective Online Convex Optimization (MO-OCO). The biggest challenge in the design of MO-OCO is the derivation of an appropriate regret definition in

the multi-objective setting. Since the multiple objectives form a vector space, we need a discrepancy metric to scalarize the loss vector. Specifically, we adopt the Pareto suboptimality gap (PSG), which is a distance-based discrepancy metric extensively used in multi-objective bandits (Turgay et al., 2018; Lu et al., 2019a). Then analogously to the single-objective online setting, we define the multi-objective static regret and the multi-objective dynamic regret. However, since PSG is a metric motivated purely from the geometric view, it is intrinsically difficult to directly optimize using first-order gradient-based iterative methods. To remedy this problem, for the multi-objective dynamic regret, via a highly non-trivial transformation, we derive its equivalent regret in the unconstrained max-min form, which is much easier to optimize. Unfortunately, we further show that such an equivalence does not hold for the multi-objective static regret since PSG always yields non-negative measurements. Hence, in this paper, we mainly focus on studying the multi-objective dynamic regret and leave the complete treatment of its static counterpart as an open problem.

Based on the proposed MO-OCO framework, we develop the Online Mirror Multiple Descent (OMMD) algorithm. The key module of OMMD is the gradient composition scheme, which utilizes the information of all the individual gradients to compute a composite gradient that tends to descend all the losses simultaneously. By directly applying the min-norm solver (Désidéri, 2012) in the offline setting to determine the composition weights, we give the first variant of OMMD termed as OMMD-I. However, the min-norm solver only uses the instantaneous gradients and ignores the historical information, which can be very unstable in the online setting where the losses and the gradients at different rounds can vary wildly. To make the learning process more stable, we introduce a carefully designed $L1$-regularizer to the vanilla min-norm solver, which results in the second variant of OMMD, namely OMMD-II.

We then give the theoretical analysis of the proposed OMMD algorithm. Specifically, we derive a non-trivial dynamic regret bound of OMMD-II, which includes that of OMMD-I as a special case. The dynamic regret bound is in the same order $O(V_T^{1/3}T^{2/3})$, where $T$ is the time horizon and $V_T$ is the temporal variability at $T$, as that of its single-objective counterpart(Zhang et al., 2018). We further show that the regret bound of OMMD-II is lower than that of OMMD-I.

To evaluate the effectiveness of our proposed algorithm, we conduct extensive experiments using both simulation datasets and real-world datasets. Specifically, we first design simulation experiments to verify the capability of OMMD-II to track the dynamic online data streams. Then we successfully realize adaptive regularization for online learning using our MO-OCO formalism, which demonstrates the effectiveness of OMMD-II in the convex setting. We further conduct online multi-task learning experiments with deep neural networks, the results of which show that both OMMD-I and OMMD-II are effective in the non-convex setting. Moreover, in both simulation and multi-task deep experiments, OMMD-II yields better performance than OMMD-I, which verifies the theoretical superiority of the regularized min-norm method over the vanilla min-norm method.

In summary, in this paper, we give the first systematic study of multi-objective online learning, which includes novel framework, algorithm design and theoretical analysis. We believe that our paper paves the way for future research on multiple-objective optimization and multi-task learning.

## 2 PRELIMINARIES

In this section, we briefly review the necessary background knowledge of online convex optimization and multi-objective optimization.

### 2.1 ONLINE CONVEX OPTIMIZATION

**Online Convex Optimization (OCO)** (Zinkevich, 2003; Hazan, 2019) is the most commonly adopted framework for designing online learning algorithms. It can be viewed as a structured repeated game between a learner and an adversary. At each round $t \in \{1, \ldots, T\}$, the learner is required to generate a decision $x_t$ from a convex compact set $\mathcal{X} \subset \mathbb{R}^n$. Then the adversary replies the learner with a convex function $f_t : \mathcal{X} \to \mathbb{R}$ and the learner suffers the loss $f_t(x_t)$. The goal of the learner is to minimize the regret with respect to the best fixed decision in hindsight, i.e.,

$$R_S(T) = \sum_{t=1}^{T} f_t(x_t) - \min_{x^* \in \mathcal{X}} \sum_{t=1}^{T} f_t(x^*).$$

Note that the above regret is the **static regret** (Hall and Willett, 2013), which compares the learner's cumulative loss with that of a fixed decision. There is another version of regret, namely the **dynamic regret** (Hall and Willett, 2013; Zhang et al., 2018), which compares the learner's cumulative loss with that of a sequence of changing decisions, i.e.,

$$R_D(T) = \sum_{t=1}^{T} f_t(x_t) - \sum_{t=1}^{T} \min_{x_t^* \in \mathcal{X}} f_t(x_t^*).$$

Any meaningful regret is required to be sublinear in $T$, i.e., $\lim_{T \to \infty} R_{S/D}(T)/T = 0$, which implies that when $T$ is large enough, the learner can perform as well as the best fixed decision in hindsight (for static regret) or the changing optimal decisions at each round (for dynamic regret).

**Online Mirror Descent (OMD)** (Hazan, 2019) is a classic first-order online learning algorithm. At each round $t \in \{1, \ldots, T\}$, OMD yields its decision using the following formula

$$x_{t+1} = \arg\min_{x \in \mathcal{X}} \eta \langle \nabla f_t(x_t), x \rangle + B_R(x, x_t),$$

where $\eta$ is the step size, $R : \mathcal{X} \to \mathbb{R}$ is the regularization function, and $B_R(x, x') = R(x) - R(x') - \langle \nabla R(x'), x - x' \rangle$ is the Bregman divergence induced from $R$. As a generic algorithm, by instantiating different regularization functions, OMD can induce two important algorithms, i.e., Online Gradient Descent (Zinkevich, 2003) and Online Exponentiated Gradient (Hazan, 2019). Several papers work on the dynamic regret of online mirror descent (Jadbabaie et al., 2015; Shahrampour and Jadbabaie, 2017).

## 2.2 MULTI-OBJECTIVE OPTIMIZATION

**Multiple-objective optimization (MOO)** is concerned with solving the problems of optimizing multiple objective functions simultaneously (Zitzler and Thiele, 1999; Sener and Koltun, 2018). In general, since different objectives may conflict with each other, there is no single solution that can optimize all the objectives at the same time. Instead, MOO seeks to find solutions that achieve Pareto optimality. In the following, we exposit Pareto optimality and related definitions more formally using a vector-valued loss $H = (h^1, \ldots, h^m)^\top$ as objectives, where $m \geq 2$ and $h^i : \mathcal{K} \to \mathbb{R}$, $i \in \{1, \ldots, m\}, \mathcal{K} \subset \mathbb{R}$, is the $i$-th loss function.

**Definition 1** (**Pareto optimality**). *(a) For any two solutions $x, x' \in \mathcal{K}$, we say that $x$ dominates $x'$, denoted as $x \prec x'$ or $x' \succ x$, if $h^i(x) \leq h^i(x')$ for all $i$, and there exists one $i$ such that $h^i(x) < h^i(x')$; otherwise, we say that $x$ does not dominate $x'$, denoted as $x \not\prec x'$ or $x' \not\succ x$.
(b) A solution $x^* \in \mathcal{K}$ is called Pareto optimal if it is not dominated by any other solution in $\mathcal{K}$.*

There may exist multiple Pareto optimal solutions. For example, it is easy to show that the optimizer of any single objective, say, $x_1^* \in \arg\min_{x \in \mathcal{K}} h^1(x)$, is Pareto optimal. Different Pareto optimal solutions reflect different trade-offs among the objectives (Sener and Koltun, 2018; Lin et al., 2019).

**Definition 2** (**Pareto front**). *(a) All Pareto optimal solutions form the Pareto set, denoted as $\mathcal{P}_\mathcal{K}(H)$.
(b) The image of $\mathcal{P}_\mathcal{K}(H)$ constitutes the Pareto front, denoted as $\mathcal{P}(H) = \{H(x) \mid x \in \mathcal{P}_\mathcal{K}(H)\}$.*

Now that we've established the notion of optimality in MOO, we proceed to introduce the metrics that measure the discrepancy of an arbitrary solution $x \in \mathcal{K}$ from being optimal. Recall that, in the single-objective setting with merely one loss function $h : \mathcal{Q} \to \mathbb{R}$, where $\mathcal{Q} \subset \mathbb{R}$, for any $z \in \mathcal{Q}$, the loss gap $h(z) - \min_{z'' \in \mathcal{Q}} h(z'')$ is directly the discrepancy measure. However, in MOO with more than one loss, for any $x \in \mathcal{K}$, the loss gap $H(x) - H(x'')$, where $x'' \in \mathcal{P}_\mathcal{K}(H)$, is a vector. Intuitively, the desired discrepancy metric shall scalarize the vector-valued loss gap and yield the value 0 for any Pareto optimal solution. In general, there are two commonly used discrepancy metrics in MOO, namely Pareto suboptimality gap (PSG) (Turgay et al., 2018) and Hypervolume (HV) (Bradstreet, 2011). As HV is a volume-based metric, it is very difficult to optimize or analyze via iterative algorithms (Zhang and Golovin, 2020). Hence in this paper, we adopt PSG, which has been extensively used in multi-objective bandits (Turgay et al., 2018; Lu et al., 2019a).

**Definition 3** (**Pareto suboptimality gap**). *For any $x \in \mathcal{K}$, the Pareto suboptimality gap to a given comparator set $\mathcal{K}^* \subset \mathcal{K}$, denoted as $\Delta(x; \mathcal{K}^*, H)$, is defined as the minimal scalar $\epsilon \geq 0$ that needs to be subtracted from all entries of $H(x)$, such that $H(x) - \epsilon \mathbf{1}$ is not dominated by any point in $\mathcal{K}^*$, where $\mathbf{1}$ denotes the all-one vector in $\mathbb{R}^m$, i.e.,*

$$\Delta(x; \mathcal{K}^*, H) = \inf_{\epsilon \geq 0} \epsilon \ \ s.t. \ \forall x'' \in \mathcal{K}^*, \exists i \in \{1, \ldots, m\}, h^i(x) - \epsilon < h^i(x'').$$

Clearly, PSG is a distance-based discrepancy metric that motivated from a purely geometric viewpoint. In practice, the comparator set $\mathcal{K}^*$ is often set to be the Pareto set $\mathcal{P}_\mathcal{K}(H)$ (Turgay et al., 2018). Then for any $x \in \mathcal{K}$, its PSG is always non-negative and equals to zero if and only if $x \in \mathcal{P}_\mathcal{K}(H)$.

**Multiple Gradient Descent Algorithm (MGDA)** is an offline first-order algorithm for MOO (Fliege and Svaiter, 2000; Désidéri, 2012). At each iteration $l \in \{1, \ldots, L\}$, where $L$ is the maximum number of iterations, it first computes the gradient $\nabla h^i(x_l)$ for each objective $i \in \{1, \ldots, m\}$, and then derive the composite gradient $g_l = \sum_{i=1}^m \lambda_l^i \nabla h^i(x_l)$ as the convex combination of these multiple gradients; it applies the composite gradient to execute the gradient descent step to update the decision, i.e., $x_{l+1} = x_l - \eta g_l$, where $\eta$ is the step size. The core module of MGDA is the determination of the weights $\lambda_l = (\lambda_l^1, \ldots, \lambda_l^m)$ for the gradient composition, which is given as

$$\lambda_l = \underset{\lambda_l \in \Delta_m}{\arg\min} \| \sum_{i=1}^m \lambda_l^i \nabla h^i(x_l)\|_2^2,$$

where $\Delta_m = \{\lambda \in \mathbb{R}^m \mid \lambda^i \geq 0, i \in \{1, \ldots, m\}$, and $\sum_{i=1}^m \lambda^i = 1\}$ denotes the probabilistic simplex in $\mathbb{R}^m$. This is a min-norm solver which finds the weights in the simplex that yields the minimum $L_2$ norm of the composite gradient. Thus MGDA is also called the *min-norm* method. Existing works (Désidéri, 2012; Sener and Koltun, 2018) have shown that MGDA is guaranteed to decrease all the objectives simultaneously until it reaches a Pareto optimal decision (under the convex setting where all $h^i$ are convex functions).

## 3 MULTI-OBJECTIVE ONLINE CONVEX OPTIMIZATION

In this section, we formally formulate the framework of multi-objective optimization in the online setting, termed Multi-Objective Online Convex Optimization (MO-OCO).

We tailor the famous online convex optimization (OCO) framework to the multi-objective setting, which can be viewed as a repeated game between an online learner and the adversarial environment. At each round $t \in \{1, \ldots, T\}$, the learner generates a decision $x_t$ from a given convex compact decision set $\mathcal{X} \subset \mathbb{R}^n$. Then the adversary replies the decision with a vectoral loss function $F_t(x)$ : $\mathcal{X} \to \mathbb{R}^m$, where its $i$-th component $f_t^i(x) : \mathcal{X} \to \mathbb{R}$ belongs to the $i$-th objective, and the learner suffers the loss $F_t(x_t) \in \mathbb{R}^m$. The goal of the learner is to generate a sequence of decisions $\{x_t \mid 1 \leq t \leq T\}$ so that the cumulative loss $\sum_{t=1}^T F_t(x_t)$ can be optimized.

Recall that, in the single-objective setting, the performance metric $R(T) = \sum_{t=1}^T f_t(x_t) - f_t(x_t^*)$, i.e., the regret, compares the actual decisions $x_t$ with some comparator $x_t^* \in \mathcal{X}$ at each round $t \in \{1, \ldots, T\}$. In general, there are two common types of regret which differ in the comparators, i.e., the static regret and the dynamic regret. For the static regret, all comparators $x_t^*, \forall t \in \{1, \ldots, T\}$ are identically set as the fixed optimal decision $x^*$ w.r.t. all losses in hindsight, i.e., $x_t^* \equiv x^* \in \arg\min_{x \in \mathcal{X}} \sum_{t=1}^T f_t(x)$. For the dynamic regret, the comparator $x_t^*$ at each round $t$ is selected as the optimal decision w.r.t. the instantaneous loss $f_t$ at that round, i.e., $x_t^* \in \arg\min_{x \in \mathcal{X}} f_t(x)$.

In analogy, in the multi-objective setting, we define the regret as $R(T) = \sum_{t=1}^T \Delta_t$, where the quantity $\Delta_t$ at each round $t$ compares the actual decisions $x_t$ with some comparator $x_t^* \in \mathcal{X}$. However, in general, no single decision can optimize all the objectives at the same time. Hence, it is reasonable to compare $x_t$ with all the Pareto optimal decisions that constitute a comparator set $\mathcal{X}_t^* \subset \mathcal{X}$. Specifically, we introduce the Pareto suboptimality gap (PSG) (Turgay et al., 2018), i.e.,

$$\Delta(x_t; \mathcal{X}_t^*, F_t) = \inf_{\epsilon \geq 0} \epsilon \ \ s.t. \ \forall x'' \in \mathcal{X}_t^*, \exists i \in \{1, \ldots, m\}, f_t^i(x_t) - \epsilon < f_t^i(x''). \tag{1}$$

Then the multi-objective regret can be defined as $R(T) = \sum_{t=1}^T \Delta(x_t; \mathcal{X}_t^*, F_t)$. Given the above definitions, we can formulate the multi-objective variants of the static and dynamic regret respectively, by using different comparator set $\mathcal{X}_t^*$ at each step. Specifically, if we set all $\mathcal{X}_t^*$ to be the Pareto set of the cumulative loss $\sum_{t=1}^T F_t$, then we can formulate a regret metric termed the *multi-objective static regret* (recall that $\mathcal{P}_X(F)$ denotes the Pareto set of $F$)

$$R_{\text{MOS}}(T) := \sum_{t=1}^T \Delta(x_t; \mathcal{X}^*, F_t), \quad \text{where } \mathcal{X}^* = \mathcal{P}_X(\sum_{t=1}^T F_t).$$

Alternatively, if we set $\mathcal{X}_t^*$ to be the Pareto set of the instantaneous loss $F_t$ at each round $t$, then we can give a regret metric termed the *multi-objective dynamic regret*

$$R_{\text{MOD}}(T) := \sum_{t=1}^{T} \Delta(x_t; \mathcal{X}_t^*, F_t), \quad \text{where } \mathcal{X}_t^* = \mathcal{P}_X(F_t), \forall t \in \{1, \ldots, T\}.$$

Recall that, PSG is a zero-order metric motivated in a purely geometric sense, namely, its calculation can be viewed as a constrained minimization problem (1) with an unknown boundary $f_t^i(x''), \forall x'' \in \mathcal{X}_t^*$. Hence, it is intrinsically complex to design a first-order algorithm to optimize PSG given its unknown constraints, not to mention the regret analysis.

Surprisingly, specific to the multi-objective dynamic regret $R_{\text{MOD}}$, we can transform it into an unconstrained max-min form as follows. The derivation utilizes Pareto optimality of $\mathcal{X}_t^*$ and is highly non-trivial, which is given in the appendix due to the space limit. The equivalent form is closely related to the dynamic regret, as it recovers the dynamic regret defined for $\lambda_t^\top F_t, t \in \{1, \ldots, T\}$ if we determine $\lambda_t$ at each round $t \in \{1, \ldots, T\}$ beforehand. Moreover, it exactly reduces to the standard dynamic regret $R_D$ in the single-objective setting.

**Proposition 1.** *The multi-objective dynamic regret has an equivalent form, i.e.,*

$$R_{\text{MOD}}(T) = \sup_{x_t^* \in \mathcal{X}_t^*, 1 \le t \le T} \inf_{\lambda_1^*, \ldots, \lambda_t^* \in \Delta_m} \sum_{t=1}^{T} (\lambda_t^{*\top} F_t(x_t) - \lambda_t^{*\top} F_t(x_t^*)), \qquad (2)$$

*where $\Delta_m$ represents the probabilistic simplex in $\mathbb{R}^m$.*

**Remark.** In the single-objective setting where $m = 1$, the probabilistic simplex $\Delta_m$ collapses into a single point $\{1\}$. Moreover, the Pareto set $\mathcal{X}_t^*$ of the scalar loss function $F_t : \mathcal{X} \to \mathbb{R}$ reduces to the optimal set $\arg\min_{x \in \mathcal{X}} F_t(x)$. Hence we have $R_{\text{MOD}}(T) = \sum_{t=1}^{T} (F_t(x_t) - \min_{x \in \mathcal{X}} F_t(x))$, which is exactly the dynamic regret $R_D(T)$ in the standard online setting.

Unfortunately, for the multi-objective static regret $R_{\text{MOS}}$, such a correspondence does not exist. Here is the reason. In $R_{\text{MOS}}$, the comparator set $\mathcal{X}^*$ is the Pareto set of the cumulative loss $\sum_{t=1}^{T} F_t$ rather than the instantaneous loss $F_t$. Hence, at some specific round $t$, the actual decision $x_t$ may Pareto dominate all decisions in $\mathcal{X}^*$ w.r.t. the instantaneous $F_t$, and so we expect the discrepancy metric $\Delta_t$ to give a negative measurement. However, PSG (as well as other commonly used discrepancy metrics such as Hypervolume) is always non-negative, so the induced $R_{\text{MOD}}$ is not aligned with $R_S$. For example, when $m = 1$, we have $R_{\text{MOS}}(T) = \sup_{x^* \in \mathcal{X}^*} \sum_{t=1}^{T} \max\{F_t(x_t) - F_t(x^*), 0\}$, which can be much looser than the static regret $R_S(T) = \sup_{x^* \in \mathcal{X}^*} \sum_{t=1}^{T} F_t(x_t) - F_t(x^*)$. Therefore, the analysis of $R_{\text{MOD}}$ is intrinsically complex if we only use existing discrepancy metrics; its analysis may require a completely new discrepancy metric $\Delta_t$ that allows to give a negative measurement.

Given the limits of existing discrepancy metrics to characterize $R_{\text{MOS}}$, in this paper, we mainly focus on the dynamic variant $R_{\text{MOD}}$. Indeed, the algorithm design and theoretical analysis w.r.t. $R_{\text{MOD}}$ are already highly non-trivial. We will leave the analysis of $R_{\text{MOS}}$ for future works.

## 4 ONLINE MIRROR MULTIPLE DESCENT

In this section, we first present the Online Mirror Multiple Descent (OMMD) algorithm, then provide theoretical analysis for it.

### 4.1 THE ALGORITHM

The protocol of OMMD is given in Algorithm 1. At each round $t$, the learner computes the gradient for each loss $\nabla f_t^i(x_t)$, then determines the weights for the composition of these multiple gradients, and finally executes an online mirror descent step using the composite gradient. For simplicity, we define the matrix form of the multiple gradients as $\nabla F_t(x_t) = [\nabla f_t^1(x_t), \ldots, \nabla f_t^m(x_t)] \in \mathbb{R}^{n \times m}$.

The core module of OMMD is the composition of the multiple gradients. In intuition, the composite gradient $g_t$ shall elaborately utilize the information of all the individual gradient. As illustrated in Preliminary (see the MGDA part), in the offline setting, there is one simple yet effective scheme,

---

**Algorithm 1** Online Mirror Multiple Descent (OMMD)

---

1: **Input:** Convex set $\mathcal{X}$, time horizon $T$, regularization parameter $\alpha$, learning rate $\eta$, regularization function $R$.
2: **Initialize:** $x_1 \in \mathcal{X}$, $\lambda_0 \in \Delta_m$.
3: **for** $t = 1, \ldots, T$ **do**
4:     Predict with $x_t$ and receive the vector loss function $F_t : \mathcal{X} \rightarrow \mathbb{R}^m$.
5:     Compute the multiple gradients $\nabla F_t(x_t) = [\nabla f_t^1(x_t), \ldots, \nabla f_t^m(x_t)] \in \mathbb{R}^{n \times m}$.
6:     Determine the weights for the gradient composition

$$\lambda_t = \arg\min_{\lambda \in \Delta_m} \|\nabla F_t(x_t)\lambda\|_2^2; \qquad \text{(OMMD-I)} \qquad (3)$$

$$\lambda_t = \arg\min_{\lambda \in \Delta_m} \|\nabla F_t(x_t)\lambda\|_2^2 + \alpha\|\lambda - \lambda_{t-1}\|_1. \qquad \text{(OMMD-II)} \qquad (4)$$

7:     Compute the composite gradient $g_t = \nabla F_t(x_t)\lambda_t$.
8:     Perform online mirror descent using the composite gradient

$$x_{t+1} = \arg\min_{x \in \mathcal{X}} \eta\langle g_t, x\rangle + B_R(x, x_t). \qquad (5)$$

9: **end for**

---

i.e., the min-norm method (Désidéri, 2012; Sener and Koltun, 2018), which computes a common descent direction that can descend all the losses simultaneously. We directly apply it to the online setting, which results in the first variant of OMMD, i.e., OMMD-I, as shown in Algorithm 1.

Despite its simplicity, OMMD-I may not yield optimal decisions in the online setting, since the min-norm method is designed in the offline setting, which does not capture the characteristics of the online setting. Specifically, the composition weights $\lambda_t$ given by the min-norm solver are determined solely by the gradients at the instantaneous round $t$, regardless of the historical information. Indeed, this makes sense in the offline setting, where the optimized loss function does not change too much. In the online setting, however, the losses at different rounds can vary wildly and so are the gradients consequently. Hence, directly applying the min-norm method will yield very different $\lambda_t$s at different rounds, which makes the learning process unstable and even fail to converge.

To fix this issue, we propose to add a regularizer $r(\lambda, \lambda_{t-1})$ to the min-norm solver when determining the composition weights $\lambda_t$, where $\lambda_{t-1}$ denotes the composition weights at the precedent round. Such a regularizer ensures that the new weights $\lambda_t$ will not move too far away from the precedent weights $\lambda_{t-1}$. In principle, $r(\lambda, \lambda_{t-1})$ can take many forms such as $L_1$-norm, $L_2$-norm and KL divergence etc. Here we use the $L_1$ norm since it aligns well with the simplex constraint of $\lambda$. This $L_1$-regularized min-norm method results in the second variant of OMMD, i.e., OMMD-II, as shown in Algorithm 1. Later we will show that OMMD-II attains a lower theoretical regret bound than OMMD-I, and it is also much more robust in experiments.

### 4.2 ANALYSIS

We first provide a general bound for OMMD, which is agnostic of the choice of $\lambda_t$ at each round.

**Assumption 1** (**Bregman divergence**). *The regularization function $R$ is 1-strongly convex with respect to a norm $\|\cdot\|$. In addition, the Bregman divergence is $\gamma$-Lipschitz continuous, i.e., $B_R(x, z) - B_R(y, z) \leq \gamma\|x - y\|, \forall x, y, z \in \mathrm{dom}R$, where $\mathrm{dom}R$ denotes the domain of the regularization function $R$ and satisfies $\mathcal{X} \subset \mathrm{dom}R \subset \mathbb{R}^n$.*

**Lemma 1.** *Suppose the diameter of the decision set $\mathcal{X}$ is bounded by $D$. Assume $F_t$ is bounded, i.e., $|f_t^i(x)| \leq F$ for any $x \in \mathcal{X}, t \in \{1, \ldots, T\}, i \in \{1, \ldots, m\}$. Then for any $\delta \in \{1, \ldots, T\}$, OMMD-I or OMMD-II with composition weights $\lambda_t$ at each round $t$ attains the following regret*

$$R_{\mathrm{MOD}}(T) \leq 2\delta \sum_{t=1}^{T-1} \sup_{x \in \mathcal{X}} |f_t^i(x) - f_{t+1}^i(x)| + 4\delta FT \sum_{t=1}^{T-1} \|\lambda_t - \lambda_{t+1}\|_1$$

$$+ \frac{\eta}{2} \sum_{t=1}^{T} \|\nabla F_t(x_t)\lambda_t\|_2^2 + \frac{\gamma D}{\eta}\lceil\frac{T}{\delta}\rceil.$$

The bound can be further simplified under the assumptions of temporal variability and Lipschitz continuity, which are commonly used in dynamic regret analysis (Besbes et al., 2015; Yang et al., 2016; Campolongo and Orabona, 2021).

**Assumption 2** (**Temporal variability**). *For each $i \in \{1, \ldots, m\}$, there exists some positive and finite $V_T$ such that $\sum_{t=1}^{T-1} \sup_{x \in \mathcal{X}} |f_t^i(x) - f_{t+1}^i(x)| \leq V_T$.*

**Assumption 3** (**Lipschitz continuity**). *For each $i \in \{1, \ldots, m\}$, there exists some positive and finite $G$ such that, the $i$-th loss $f_t^i$ at each round $t \in \{1, \ldots, T\}$ is $G$-Lipschitz continuous w.r.t. $\|\cdot\|$, i.e., $|f_t^i(x) - f_t^i(x')| \leq G\|x - x'\|$.*

In the convex setting, Lipschitz continuity guarantees bounded gradients, i.e., $\|\nabla f_t^i(x)\|_* \leq G$ for any $t \in \{1, \ldots, T\}, i \in \{1, \ldots, m\}, x \in \mathcal{X}$. We can now derive a general regret bound for OMMD.

**Theorem 1.** *Assume the step size $\eta$ satisfies $\frac{4V_T}{G^2T} \leq \eta \leq \frac{4V_T}{G^2}$. Then OMMD-I or OMMD-II with weights $\lambda_t$ at each round $t$ attains the following regret*

$$R_{\text{MOD}}(T) \leq \frac{\eta G^2 T}{2} + \frac{\eta}{2} \sum_{t=1}^{T} (\|\nabla F_t(x_t)\lambda_t\|_2^2 + \frac{8FG^2T^2}{V_T}\|\lambda_t - \lambda_{t-1}\|_1) + \frac{4\gamma DV_T}{\eta^2 G^2}.$$

**Remark.** The above theorem shows the **theoretical superiority of OMMD-II** over OMMD-I or naive linear scalarization that always uses fixed composition weights $\lambda_t \equiv \lambda$ at each round. In particular, with the proper choice of $\alpha = \frac{FG^2T^2}{V_T}$, OMMD-II adaptively selects $\lambda_t$ to minimize the term $\|\nabla F_t(x_t)\lambda_t\|_2^2 + \alpha\|\lambda_t - \lambda_{t-1}\|_1$. This term will become larger with any other choice of $\lambda_t$.

Finally, specific to OMMD-II, we have the following regret bound. Note that, this bound actually relies on the assumption that $\Omega(1) \leq V_T \leq o(T)$, which is implicitly assumed in other works for dynamic online learning (Besbes et al., 2015; Yang et al., 2016; Campolongo and Orabona, 2021). We also extend the bound to arbitrary value of $V_T \geq 0$ (see Corollary 2 in the appendix).

**Corollary 1.** *With $\eta = \frac{2}{G}(\frac{\gamma DV_T}{GT})^{1/3}$ and $\alpha = \frac{8FG^2T^2}{V_T}$, OMMD-II achieves the following regret*

$$R_{\text{MOD}}(T) \leq O(V_T^{1/3}T^{2/3}).$$

## 5 EXPERIMENTS

In this section, we conduct extensive simulation and real-world experiments to evaluate the effectiveness of our proposed algorithm. In our experiments, we mainly compare our proposed OMMD-II with two baseline algorithms: (i) *linear optimal (lin-opt)* performs single-objective online learning using the linearized loss $\lambda^\top F$ at each round $t$, where the linearization weights $\lambda$ is fixed and decided by a grid search on $\Delta_m$; note that, it is equivalent to using a fixed composition weight $\lambda_t = \lambda$ in OMMD. (ii) *min-norm* extends the famous min-norm method (Désidéri, 2012; Sener and Koltun, 2018) to the multi-objective online setting, which has been described as OMMD-I in Algorithm 1.

### 5.1 SIMULATION EXPERIMENTS: TRACKING THE PARETO FRONT

In the simulation setup, the goal is to track two points $\xi_t^1, \xi_t^2$ moving on the plane $\mathbb{R}^2$. The two points cycle along a circle with a ratius of 1, i.e., $\mathcal{C} = \{\xi \in \mathbb{R}^2 \mid \|\xi\|_2 = 1\}$, namely, for each $i \in \{1, 2\}$, $\xi_t^i = (\cos\theta_t^i, \sin\theta_t^i)$ is determined by some angle $\theta_t^i$. For each $i \in \{1, 2\}$, we set a positive integer $P^i$ as the rotating period, which is unknown to the learner. The two points are initialized by $\theta_1^1 = 0$ and $\theta_1^2 = \pi/2$, and iteratively generated as follows: at each round $t$, for each $i \in \{1, 2\}$, the adversary independently samples an angle $\delta_t^i$ from a Gaussian distribution $\mathcal{N}(2\pi/P^i, 1/\sqrt{P^i})$, then generates the next point $\xi_{t+1}^i$ via $\theta_{t+1}^i = \theta_t^i - \delta_t^i$. Note that, we have $\mathbb{E}\theta_{t+1}^i = \theta_1^i + 2\pi t/P^i$, which implies that in average $\xi_t^i$ rotates clockwise uniformly along the circle $\mathcal{C}$, with a period of $P^i$.

At each round $t$, the learner generates a decision $x_t$ from $\mathcal{X} = \{x \in \mathbb{R}^2 \mid \|x\|_2 \leq 2\}$, which is a L2-norm ball with a radius of 2. Then it acquires the positions of $\xi_t^1, \xi_t^2$ and suffer two losses $f_t^i(x_t) = \|x_t - \xi_t^i\|_2/2$ for $i \in \{1, 2\}$. In this problem, the Pareto set of the vectoral loss $f_t = (f_t^1, f_t^2)$ at each round $t$ is exactly the line segment linking $\xi_t^1$ and $\xi_t^2$, i.e., $\mathcal{X}_t^* = \{\lambda\xi_t^1 + (1-\lambda)\xi_t^2 \mid \lambda \in [0,1]\}$. For any decision $x_t \in \mathcal{X}$, its PSG exactly equals to the squared distance between $x_t$ and the line segment. The setup and PSG measurement are summarized in Figure 1 (left plot).

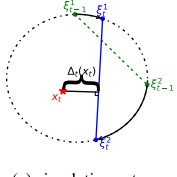 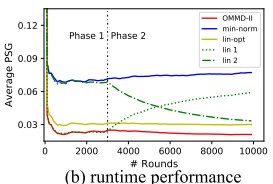 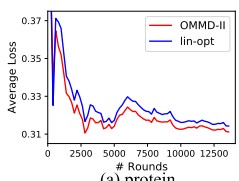 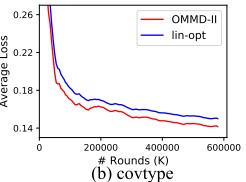

(a) simulation setup    (b) runtime performance

(a) protein    (b) covtype

Figure 1: Simulation setup and results. *Left:* Target points $\xi_t^1, \xi_t^2$ cycle along the circle. At each round $t$, the Pareto set is the line segment $[\xi_t^1, \xi_t^2]$, and PSG measures the distance between $x_t$ and the line segment. *Right:* Performance comparison of OMMD-II and baselines.

Figure 2: Results to verify the effectiveness of adaptive regularization using OMMD-II in real-world online classification tasks. The two plots compare the performance of adaptive regularization (OMMD-II) and fixed regularization (*lin-opt*) on *protein* and *covtype* datasets.

In our experiments, we set $T = 10,000$. To simulate the adversarial dynamic change, we further set $P^1 = 10, P^2 = 20$ for the first $T_1 = 3,000$ rounds, and $P^1 = 20, P^2 = 10$ for the remaining $T_2 = 7,000$ rounds. As our goal is the track the Pareto front, we adopt the average PSG, namely $\sum_{t \in [T]} \Delta_t(x_t)/T$, as the performance metric. Accompanying the *lin-opt* baseline, we also consider its two variants: *lin-1* which sets $\lambda$ as the optimal weights for the first $T_1$ rounds, and *lin-2* which set $\lambda$ to be optimal regarding the last $T_2$ rounds. The learning rates $\eta$ in all algorithms and the regularization parameter $\alpha$ in OMMD-II are set as what the corresponding theories suggest.

Figure 1 (right plot) shows the performance of all the examined algorithms. From the results, we observe that OMMD-II achieves the lowest PSG, showing its ability to track the Pareto front; meanwhile, *min-norm* appears very unstable in the dynamic setting, even worse than linear scalarization. In particular, by comparing OMMD-II and *lin-1*, we find that linear scalarization with carefully selected weights may attain performance comparable to our algorithm during some certain phase; however, in the dynamic setting where the pattern drifts over time, the performance of linear scalarization may become very unstable or even drop severely, while our algorithm is much more robust.

## 5.2 ONLINE CONVEX EXPERIMENTS: AN APPLICATION TO ADAPTIVE REGULARIZATION

In many real-world online applications, regularization is often applied to avoid overfitting. A most common way is to add a regularization term $r(x)$ to the loss $f_t(x)$ in each round, and optimize the regularized loss $f_t(x) + \sigma r(x)$ instead (McMahan, 2011). The strength $\sigma$ of regularization is often treated as a hyperparameter that needs to be decided carefully beforehand, e.g., via a grid search. The formalism of multi-objective online learning provides an another way to realize regularization. Since $r(x)$ often measures the complexity of $x$, it can be viewed as the second objective to be optimized. Specifically, by constructing a vectoral loss $F_t(x) = (f_t(x), r(x))$ in each round, we can cast the regularized online learning into a two-objective online optimization problem. Compared to the previous approach with a fixed strength $\sigma$, in our approach, the strength is implied by the weights in gradient composition, namely $\sigma_t = \lambda_t^2 / \lambda_t^1$, which is adaptive at each round $t$.

We run experiments on two online benchmark datasets. (i) *protein*: A bioinformatics dataset for protein type classification (Wang, 2002), which has 17 thousand instances with 357 features. (ii) *covtype*: A biological dataset collected from a non-stationary environment, whose goal is to predict the cover type of forests at a particular location (Blackard and Dean, 1999), which has 50 thousand instances with 54 features. For both classification tasks, we use the logistic loss as the first objective, and the squared $L2$-norm of model parameters (i.e., the $L2$-regularizer) as the second objective.

In the experiments, we adopt $L2$-norm balls as the decision set and set the diameter $K = 10$. For OMMD-II, the parameter $\alpha$ is simply set as 0.2. For fixed regularization, the strength $\sigma = \lambda_t^2 / \lambda_t^1$ is determined by a grid search over $(\lambda_t^1, \lambda_t^2) \in \Delta_2$, and we denote this method as *lin-opt*. Moreover, for both OMMD-II and *lin-opt*, learning rates $\eta$ are decided via a grid search over $\{0.1, 0.2, \ldots, 2.0\}$.

Since the ultimate goal of regularization is to enhance predictive performance, we adopt the average of cumulative loss, namely $\sum_{t \in [T]} l_t(x_t)/T$ where $l_t(x_t)$ is the classification loss at round $t$, as the performance metric. The performance of both algorithms are reported in Figure 2. The results show that OMMD-II attains lower loss than *lin-opt* in all the examined tasks, which shows the superiority of adaptive regularization using our online MOO technique over fixed regularization.

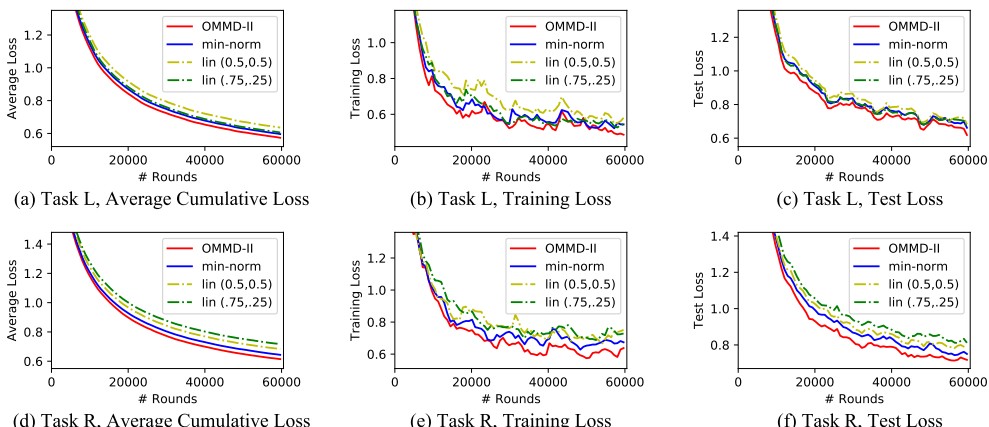

| (a) Task L, Average Cumulative Loss | (b) Task L, Training Loss | (c) Task L, Test Loss |
| (d) Task R, Average Cumulative Loss | (e) Task R, Training Loss | (f) Task R, Test Loss |

Figure 3: Results to verify the effectiveness of OMMD-II for online deep learning. The plots show the average cumulative loss, training loss and test loss for both tasks (*task L/R*) on MultiMNIST.

### 5.3 ONLINE NON-CONVEX EXPERIMENTS: MULTI-TASK DEEP LEARNING

Finally, we evaluate OMMD-II in the online non-convex setting. We choose to use the MultiMNIST dataset (Sabour et al., 2017), which is a multi-objective version of the famous MNIST dataset for image classification and commonly used in deep multi-task learning (Sener and Koltun, 2018; Lin et al., 2019). In MultiMNIST, each sample is constructed by putting a random image at the top-left and another image at the bottom-right. Our goal to classify the digit on the top-left (*task L*) and to classify the digit on the bottom-right (*task R*) at the same time.

We follow Sener and Koltun (2018)'s setup and adopt the LeNet architecture. For linear scalarization, we consider two choices of weights, namely $(0.5, 0.5)$ and $(0.75, 0.25)$. For all the examined algorithms, the learning rates $\eta$ are selected via a grid search over $\{0.0001, 0.001, 0.01, 0.1\}$. For OMMD-II, the parameter $\alpha$ is set according to our theory. Note that, in online experiments, the sample arrives one after another in a sequential manner, which is different from stochastic optimization where sample batches are randomly sampled from the whole training set (Sener and Koltun, 2018).

Figure 3 compares the average cumulative loss, training loss and test loss of all the examined algorithms for both tasks. Note that, the first metric is typically used in online experiments and the last two are commonly used in stochastic experiments (Reddi et al., 2018). The results show that OMMD-II outperforms OMMD-I (*min-norm*) and linear scalarization (*lin*) in all metrics, which shows that our proposed algorithm is also effective in the online non-convex setting.

## 6 CONCLUSIONS

In this paper, we conduct a systematic study of multi-objective optimization in the online setting. We first formulate the framework of Multi-Objective Online Convex Optimization. Then we devise the Online Mirror Multiple Descent algorithm, which is the first gradient-based multi-objective online learning algorithm and has a special design when tailoring multiple gradient algorithm to online learning, namely regularized min-norm solver. Theoretically, we provide the first paradigm of regret analysis for multi-objective online convex optimization. We finally conduct extensive experiments to demonstrate the effectiveness of our proposed algorithm. Future works may include developing a framework based on the Hypervolume metric, or giving an analysis of multi-objective static regret.

## 7 ETHICS STATEMENT

As a study on a general learning problem, our work will not incur ethical issues by itself. However, ethical issues may arise if our learning method is improperly applied to some application fields - just as any other general learning method if it is misused.

## 8 REPRODUCIBILITY STATEMENT

For every theoretical statement in our paper (including proposition, lemma, theorem and corollary), we give a detailed proof in Appendix C. The codes to reproduce our empirical results are provided in the supplementary materials.

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

APPENDIX

The appendix is organized as follows. Appendix A provides more details of OMMD, including a discussion on the lazy version of mirror descent and how to efficiently compute the composition weights for the regularized min-norm solver. Appendix B supplements the theoretical results with more general regret bounds for arbitrary temporal variability $V_T \in [0, \infty)$; such a general setting is often not considered in a typical analysis of dynamic online learning (Besbes et al., 2015; Zhang et al., 2018), but we believe that adding such general bounds will make the analysis more self-contained. Appendix C provides detailed proofs of all theoretical claims in this work, which have been omitted in our main paper due to the space limit.

## A   MORE DETAILS OF THE ALGORITHM

In this section, we provide more details to help better understand the design of OMMD.

First, we note that, our proposed OMMD is actually based on the agile version of online mirror descent (Hazan, 2019), where the updated model directly moves to its projecting point onto the decision set at each round. However, we can easily make an analogy and devise a lazy version of OMMD. Specifically, we only need to use a lazy projection operation in the mirror descent step with the composite gradient instead of the agile projection operation (line 7 in Algorithm 1). Note that, the analysis of the lazy version is very similar to that of the agile version (Hazan, 2019).

Next, we show that, when calculating the weights $\lambda_t$ for gradient composition, our regularized min-norm solver only need very light computation, just as the original min-norm solver in stochastic optimization (Sener and Koltun, 2018; Lin et al., 2019). Specifically, similar to (Sener and Koltun, 2018), we first consider the setting of two objectives, namely $m = 2$. In this case, when $\lambda, \lambda_{t-1} \in \Delta_2$, the L1 regularizer $\|\lambda - \lambda_{t-1}\|_1$ equals to $2|\lambda^1 - \lambda_{t-1}^1|$. Then the optimization problem on $\lambda$ at round $t$ reduces to (since there are only two objectives, the superscripts of $\lambda^1$ and $\lambda_{t-1}^1$ are omitted)

$$\min_{\lambda \in [0,1]} \|\lambda g_1 + (1 - \lambda)g_2\|^2 + 2\alpha|\lambda - \lambda_{t-1}|.$$

Interestingly, we find that, the above problem also has an closed-form solution as below.

**Proposition 2.** *Set* $\lambda_L = (g_2^\top(g_2 - g_1) + \alpha)/\|g_2 - g_1\|^2$, *and* $\lambda_R = (g_2^\top(g_2 - g_1) - \alpha)/\|g_2 - g_1\|^2$. *Then the above optimization problem has a closed-form solution, i.e.,*

$$\lambda = \begin{cases} \max\{0, \lambda_L\}, & \lambda_L \leq \lambda_{t-1}; \\ \min\{1, \lambda_R\}, & \lambda_R \geq \lambda_{t-1}; \\ \lambda_{t-1}, & otherwise. \end{cases}$$

*Proof.* We solve the following two quadratic sub-problems respectively, namely,

$$\min_{\lambda \in [0, \lambda_{t-1}]} \|\lambda g_1 + (1 - \lambda)g_2\|^2 + 2\alpha(\lambda_{t-1} - \lambda),$$

as well as

$$\min_{\lambda \in [\lambda_{t-1}, 1]} \|\lambda g_1 + (1 - \lambda)g_2\|^2 + 2\alpha(\lambda - \lambda_{t-1}).$$

The former problem produces $\lambda = \max\{\min\{\lambda_L, \lambda_{t-1}\}, 0\}$, and the latter one gives $\lambda = \max\{\min\{\lambda_R, 1\}, \lambda_{t-1}\}$. Therefore, the solution to the original optimization problem can then be derived by comparing the optimal values of the two sub-problems, which gives the desired solution in the proposition. ∎

Now that we have derived the closed-form solution to the regularized min-norm solver with any two gradients, in priciple, we can apply Sener and Koltun (2018)'s technique to efficiently compute the solution to the solver with more than two gradients. Specifically, we can adopt the Frank-Wolfe method, namely, iteratively choose a pair of gradients and update the composition weights with these two gradients.

## B    MORE DETAILS OF THEORETICAL ANALYSIS

In this section, we provide more details and highlight technical lemmas in theoretical analysis, which are omitted in our main paper due to space limitation.

### B.1    A MORE DETAILED REGRET BOUND FOR OMMD-II

In Corollary 1, we only give the order of OMMD-II's regret bound w.r.t. the core factors $m, T$ and $V_T$. In the following theorem, we supplement a more detailed regret bound for OMMD-II which explicates the dependency of other factors on the regret. The proof is presented in Appendix C.

**Theorem 2.** *Set* $\eta = \frac{2}{G}(\frac{\gamma D V_T}{GT})^{1/3}$ *and* $\alpha = \frac{8FG^2T^2}{V_T}$. *Then OMMD-II attains the following regret*

$$R_{\text{MOD}}(T) \leq (\frac{\gamma D}{G^4})^{1/3}\frac{(V_T)^{1/3}}{T^{1/3}}\sum_{t=1}^{T}\inf_{\lambda \in \Delta_m}(\|\nabla F_t(x_t)\lambda\|_*^2 + \frac{8FG^2T^2}{V_T}\|\lambda - \lambda_{t-1}\|_1)$$
$$+ 2(\gamma D G^2)^{1/3}(V_T)^{1/3}T^{2/3}.$$

Note that, as we have discussed in our main paper, just like its simplified version Corollary 1, the above theorem is derived under the assumption of "regular" temporal variability, namely $\Omega(1) \leq V_T \leq o(T)$, which is implicitly assumed in other works for dynamic online learning (Besbes et al., 2015; Yang et al., 2016; Campolongo and Orabona, 2021). To give a more general analysis, in the following subsection, we also provide strict regret bounds that are valid for arbitrary $V_T \in [0, \infty)$.

### B.2    REGRET BOUNDS FOR OMMD-II UNDER ARBITRARY TEMPORAL VARIABILITY

We here provide a more strict regret bound for OMMD-II under arbitrary temporal variability $V_T \in [0, \infty)$. Specifically, we first give a detailed regret bound in analogy to Theorem 2. The proof of this theorem is given in Appendix C.

**Theorem 3.** *In OMMD-II, set* $\eta = \min\{\max\{\frac{2}{G}(\frac{\gamma D V_T}{GT})^{1/3}, \frac{4V_T}{G^2T}\}, \frac{4V_T}{G^2}\}$ *and* $\alpha = \frac{8FG^2T^2}{V_T}$. *Then it attains the regret bound*

$$R_{\text{MOD}}(T) \leq \max\{(\frac{\gamma D}{G^4})^{1/3}\frac{(V_T)^{1/3}}{T^{1/3}}\sum_{t=1}^{T}\inf_{\lambda \in \Delta_m}\left(\frac{8FG^2T^2}{V_T}\|\lambda - \lambda_{t-1}\|_1 + \|\nabla F_t(x_t)\lambda\|_*^2\right)$$
$$+ 2(\gamma D G^2)^{1/3}(V_T)^{1/3}T^{2/3}, \quad 6V_T, \quad \frac{3G}{2}(2\gamma D)^{1/2}T^{1/2}\}.$$

The above bound can be rewritten into a simpler form, if we are only interested in its order w.r.t. $m, T$ and $V_T$, just as in Corollary 1.

**Corollary 2.** *In OMMD-II, set* $\eta = \min\{\max\{\frac{2}{G}(\frac{\gamma D V_T}{GT})^{1/3}, \frac{4V_T}{G^2T}\}, \frac{4V_T}{G^2}\}$ *and* $\alpha = \frac{8FG^2T^2}{V_T}$. *Then it attains the following regret*

$$R_{\text{MOD}}(T) \leq \max\{O(V_T^{1/3}T^{2/3}), O(V_T), O(T^{1/2})\}.$$

## C    OMITTED PROOFS

In this section, we provide the detailed proofs of Proposition 1, Lemma 1, Theorem 1 and Corollary 1 in our main paper as well as Theorem 2, Theorem 3 and Corollary 2 in the appendix.

### C.1    PROOF OF PROPOSITION 1

*Proof.* We first analyze the PSG measurement $\Delta_t(x_t)$ at each round $t \in \{1, \ldots, T\}$. Specifically, for any comparator $x \in \mathcal{X}$, we first define the **pair-wise suboptimality gap** between decisions $x_t$ and $x$, namely,

$$\delta_t(x_t; x) = \inf_{\epsilon \geq 0}\{\epsilon \mid F_t(x_t) - \epsilon\mathbf{1} \not\succ F_t(x)\}.$$

Then $\Delta_t(x_t)$ can be evaluated via the pair-wise suboptimality gap as $\Delta_t(x_t) = \sup_{x \in \mathcal{P}_X(F_t)} \delta_t(x_t, x)$.

We then focus on the pair-wise gap $\delta_t(x_t; x)$ w.r.t. any Pareto optimal decision $x \in \mathcal{X}_t^* \equiv \mathcal{P}_X(F_t)$. Since $x$ is a Pareto optimal decision of $F_t$, from the definition of Pareto optimality, there must exist some $i \in \{1, \ldots, m\}$ such that $f_t^i(x_t) - f_t^i(x) \geq 0$.

The pair-wise suboptimality gap has an equivalent expression, namely,

$$\delta_t(x_t; x) = \min_{k \in \{1, \ldots, m\}} \{(f_t^k(x_t) - f_t^k(x))_+\},$$

where $(l)_+ = \max\{l, 0\}, l \in \mathbb{R}$ is the truncation operator. Denote $\mathcal{U}_m = \{e_k \mid 1 \leq k \leq m\}$ as the set of all unit vector in $\mathbb{R}^m$, then we equivalently have

$$\delta_t(x_t; x) = \min_{\lambda \in \mathcal{U}_m} \lambda^\top (F_t(x_t) - F_t(x))_+.$$

Note that, we here slightly abuse the truncation operator $(l)_+$ to allow $l$ to be a vector in $\mathbb{R}^m$, which represents the vector whose $i$-th coordinate equals to $\max\{l^i, 0\}$ for any $i \in \{1, \ldots, m\}$.

Now the calculation of $\delta_t(x_t; x)$ becomes a minimization problem over $\lambda \in \mathcal{U}_m$. Since $\mathcal{U}_m$ is a discrete set, we can apply a linear relaxation trick. Specifically, we now turn to minimize the quantity $p(\lambda) = \lambda^\top (F_t(x_t) - F_t(x))_+$ over the convex curvature of $\mathcal{U}_m$, which is exactly the probability simplex $\Delta_m = \{\lambda \in \mathbb{R}^m \mid \lambda \succ \mathbf{0}, \|\lambda\|_1 = 1\}$. Note that, $\mathcal{U}_m$ contains all vertexes of $\Delta_m$. Since $\inf_{\lambda \in \Delta_m} p(\lambda)$ is a linear optimization problem, the minimal point ${\lambda_t^*}^\top$ must be a vertex of the simplex, i.e., ${\lambda_t^*}^\top \in \mathcal{U}_m$. Thus the relaxed problem is equivalent to the original problem, namely,

$$\min_{\lambda \in \mathcal{U}_m} \lambda^\top (F_t(x_t) - F_t(x))_+ = \inf_{\lambda \in \Delta_m} \lambda^\top (F_t(x_t) - F_t(x))_+.$$

For now, we have transformed the calculation of pair-wise suboptimality gap $\delta_t(x_t; x)$ into a optimization problem of finding the minimal linear scalarization of $(F_t(x_t) - F_t(x))_+$. Hence, the PSG at each round $t$ can be expressed as

$$\Delta_t(x_t) = \sup_{x \in \mathcal{X}_t^*} \inf_{\lambda \in \Delta_m} \lambda^\top (F_t(x_t) - F_t(x))_+.$$

In the above expression, the existence of truncation operator $(\cdot)_+$ will incur irregularity (i.e., non-linearity) when we try to optimize the loss $F_t$. Suprisingly, we find that such operator **can be dropped** when we compute $\Delta_t(x_t)$ w.r.t. the Pareto set $\mathcal{X}_t^*$, as shown in the following.

**Lemma 2.** *Let $\mathcal{X}_t^*$ be the Pareto set of $F_t : \mathcal{X} \to \mathbb{R}^m$. Then for any $x_t \in \mathcal{X}$, it holds that*

$$\sup_{x \in \mathcal{X}^*} \inf_{\lambda \in \Delta_m} \lambda^\top (F_t(x_t) - F_t(x))_+ = \sup_{x \in \mathcal{X}_t^*} \inf_{\lambda \in \Delta_m} \lambda^\top (F_t(x_t) - F_t(x)).$$

*Proof.* Define the alternative "gap" metric without the truncation operator as $\delta_t'(x_t; x) = \inf_{\lambda \in \Delta_m} \lambda^\top (F_t(x_t) - F_t(x))$. Moreover, define the supremum of $\delta_t'(x_t; x)$ over $x \in \mathcal{X}_t^*$ as $\Delta_t'(x_t) = \sup_{x \in \mathcal{X}_t^*} \delta_t'(x_t; x)$. Then from the definition of truncation operator $(\cdot)_+$, we have $\delta(x_t; x) \geq \delta'(x_t; x)$ and $\Delta_t(x_t) \geq \Delta_t'(x_t)$.

It then suffices to prove that, for any given $x_t \in \mathcal{X}$, there exists some certain $x_t^* \in \mathcal{X}_t^*$ such that the value of $\delta'(x_t; x_t^*)$ can be as large as $\Delta_t(x_t)$. Indeed, if this is the case, then $\Delta_t(x_t) \leq \Delta_t'(x_t)$, and hence the two quantities $\Delta_t(x_t)$ and $\Delta_t'(x_t)$ are equal. We consider the following two cases:

(i) $x_t$ is already a Pareto optimal point of $F_t$, i.e., $x_t \in \mathcal{X}_t^*$. Then from the definition of PSG, we directly have $\Delta_t(x_t) = 0$. Notice that $\delta'(x_t; x_t) = 0$, and hence $\Delta_t(x_t) = \sup_{x \in \mathcal{X}_t^*} \delta'(x_t; x) \geq \delta'(x_t; x_t) = 0$. Consequently, the relation $\Delta_t(x_t) \leq \Delta_t'(x_t)$ holds in this case.

(ii) $x_t$ is not a Pareto optimal point of $F_t$, i.e., $x_t \notin \mathcal{X}_t^*$. Then we have $\Delta_t(x_t) > 0$. Set $\epsilon = \Delta_t(x_t)$, and denote $x_t^* \in \arg\max_{x \in \mathcal{X}_t^*} \delta_t(x_t; x)$, then $\delta_t(x_t; x_t^*) = \epsilon > 0$. Therefore, from the definition of $\delta_t(x_t; x_t^*)$ we know that, for any $i \in \{1, \ldots, m\}$, we have $f_t^i(x_t) - f_t^i(x_t^*) \geq \epsilon$. Thus all entries of $F_t(x_t) - F_t(x^*)$ are positive, and we have $(F_t(x_t) - F_t(x^*))_+ = F_t(x_t) - F_t(x^*)$. Consequently, we have $\Delta_t'(x_t) = \sup_{x \in \mathcal{X}_t^*} \delta_t'(x_t; x) \geq \delta'(x_t; x_t^*) = \delta(x_t; x_t^*) = \Delta_t(x_t)$. ∎

From the above lemma, the PSG measurement at round $t$ has an equivalent form as

$$\Delta_t(x_t) = \sup_{x \in \mathcal{X}_t^*} \inf_{\lambda \in \Delta_m} \lambda^\top (F_t(x_t) - F_t(x)),$$

and correspondingly, the multi-objective dynamic regret becomes

$$R_{\text{MOD}}(T) = \sum_{t=1}^{T} \Delta_t(x_t) = \sum_{t=1}^{T} \sup_{x \in \mathcal{X}_t^*} \inf_{\lambda \in \Delta_m} \lambda^\top (F_t(x_t) - F_t(x)).$$

Since the time horizon $T$ is finite, we can first swap the summation over $t$ and the supremum over $x$, then swap the summation over $t$ and the infimum over $\lambda$. Then the multi-objective dynamic regret further equals to

$$R_{\text{MOD}}(T) = \sup_{x_t^* \in \mathcal{X}_t^*, 1 \le t \le T} \inf_{\lambda_1^* \dots, \lambda_t^* \in \Delta_m} \sum_{t=1}^{T} (\lambda_t^{*\top} F_t(x_t) - \lambda_t^{*\top} F_t(x_t^*)),$$

which proves the proposition. ■

## C.2 PROOF OF LEMMA 1

*Proof.* For OMMD whose the composition weights are $\lambda_t$ at round $t$, before analyzing its regret bound, we first introduce two useful lemmas.

**Lemma 3.** *For OMD with stepsize $\eta$ and loss $\lambda_t^\top F_t(x)$, we have the following recursion*

$$\lambda_t^\top F_t(x_t) - \lambda_t^\top F_t(x_t^*) \le \frac{1}{\eta}(B_R(x_t^*; x_t) - B_R(x_t^*; x_{t+1})) + \frac{\eta}{2}\|\nabla F_t(x_t)\lambda_t\|_*^2, \qquad (6)$$

*for any $t \in \{1, \dots, T\}$.*

*Proof.* Our proof is similar to the analysis of OMD in the single-objective setting (Srebro et al., 2011; Cesa-Bianchi et al., 2012). Specifically, fix $f_t = \lambda_t^\top F_t$ and $g_t = \lambda_t^\top F_t(x_t)$. From the convexity of $f_t$, we have

$$f_t(x_t) - f_t(x_t^*) \le g_t^\top(x_t - x_t^*) = g_t^\top(x_{t+1} - x_t^*) + g_t^\top(x_t - x_{t+1}).$$

From the first-order optimal condition of $x_{t+1}$, for any $x' \in \mathcal{X}$, we have

$$(\eta \nabla F_t(x_t)\lambda_t + \nabla R(x_{t+1}) - \nabla R(x_t))^\top(x' - x_{t+1}) \ge 0.$$

We set $x' = x_t^*$ in the above inequality, and consequently derive

$$f_t(x_t) - f_t(x_t^*) \le \frac{1}{\eta}(\nabla R(x_{t+1}) - \nabla R(x_t))^\top(x_t^* - x_{t+1}) + g_t^\top(x_t - x_{t+1}).$$

Recall the definition of Bregman divergence $B_R$. We can check that (also see (Beck and Teboulle, 2003))

$$B_R(x_t^*, x_t) - B_R(x_t^*, x_{t+1}) - B_R(x_{t+1}, x_t) = (\nabla R(x_{t+1}) - \nabla R(x_t))^\top(x_t^* - x_{t+1}).$$

Since $R$ is 1-strongly convex, we have $B_R(x_{t+1}, x_t) \ge \|x_{t+1} - x_t\|^2/2$. Hence

$$f_t(x_t) - f_t(x_t^*) \le \frac{1}{\eta}(B_R(x_t^*, x_t) - B_R(x_t^*, x_{t+1}) - \frac{1}{2}\|x_{t+1} - x_t\|^2) + g_t^\top(x_t - x_{t+1}).$$

Moreover, from the Cauchy-Schwartz inequality we have

$$g_t^\top(x_t - x_{t+1}) \le \frac{\eta}{2}\|g_t\|_*^2 + \frac{1}{2\eta}\|x_t - x_{t+1}\|^2.$$

Combining the above two inequalities, we can prove the lemma. ■

**Lemma 4.** *For an arbitrary comparator sequence $u_1, \dots, u_T \in \mathcal{X}$, we have*

$$\sum_{t=1}^{T} \lambda_t^\top (F_t(x_t) - F_t(u_t)) \le \frac{\gamma}{\eta} \sum_{t=1}^{T-1} \|u_t - u_{t+1}\| + \frac{1}{\eta} B_R(u_1, x_1) + \frac{\eta}{2} \sum_{t=1}^{T} \|\nabla F_t(x_t)\lambda_t\|_*^2$$

*Proof.* Summing the inequality in Lemma 3 over $t \in \{1, \ldots, T\}$, we have

$$
\begin{aligned}
\sum_{t=1}^{T} \lambda_t^\top (F_t(x_t) - F_t(u_t)) &\leq \sum_{t=1}^{T} \frac{1}{\eta} (B_R(u_t, x_t) - B_R(u_t, x_{t+1})) + \frac{\eta_t}{2} \|\nabla F_t(x_t) \lambda_t\|_*^2 \\
&\leq \frac{1}{\eta} \sum_{t=1}^{T-1} (B_R(u_{t+1}, x_{t+1}) - B_R(u_t, x_{t+1})) + \frac{1}{\eta} B_R(u_1, x_1) + \frac{\eta_t}{2} \|\nabla F_t(x_t) \lambda_t\|_*^2.
\end{aligned}
$$

Recall the assumption that $B_R(x, z) - B_R(y, z) \leq \gamma \|x - y\|, \forall x, y, z \in \mathcal{X}$. We then obtain

$$
\sum_{t=1}^{T} \lambda_t^\top (F_t(x_t) - F_t(u_t)) \leq \frac{1}{\eta} \sum_{t=1}^{T-1} \gamma \|u_t - u_{t+1}\| + \frac{1}{\eta} B_R(u_1, x_1) + \frac{\eta}{2} \|\nabla F_t(x_t) \lambda_t\|_*^2,
$$

which proves the lemma. ∎

We can now return to prove Lemma 1. Notice that, for an arbitrary comparator sequence $u_1, \ldots, u_T \in \mathcal{X}$, the Pareto tracking regret can be decomposed as

$$
\sum_{t=1}^{T} \lambda_t^\top F_t(x_t) - \sum_{t=1}^{T} \lambda_t^\top F_t(x_t^*) = \underbrace{\sum_{t=1}^{T} \lambda_t^\top F_t(x_t) - \sum_{t=1}^{T} \lambda_t^\top F_t(u_t)}_{\text{term } A} + \underbrace{\sum_{t=1}^{T} \lambda_t^\top F_t(u_t) - \sum_{t=1}^{T} \lambda_t^\top F_t(x_t^*)}_{\text{term } B}.
$$

We now instantiate the comparator sequence to be a piece-wise stationary sequence, i.e.,

$$
\{u_1, \ldots, u_T\} = \Big\{ \underbrace{w_{\mathcal{I}_1}^\star, \ldots, w_{\mathcal{I}_1}^\star}_{\Delta \text{ times}}, \ldots, \underbrace{w_{\mathcal{I}_{\lceil \frac{T}{\Delta}\rceil-1}}^\star, \ldots, w_{\mathcal{I}_{\lceil \frac{T}{\Delta}\rceil-1}}^\star}_{\Delta \text{ times}}, \underbrace{w_{\mathcal{I}_{\lceil \frac{T}{\Delta}\rceil}}^\star, \ldots, w_{\mathcal{I}_{\lceil \frac{T}{\Delta}\rceil}}^\star}_{(1+\frac{T}{\Delta}-\lceil \frac{T}{\Delta}\rceil)\Delta \text{ times}} \Big\},
$$

which starts with $w_1^*$ and only changes for every $\Delta$ steps ($\Delta$ is an integer such that $\Delta \leq T$). More specifically, for any $i \in \{1, \ldots, \lceil T/\Delta\rceil\}$, denote $p_i = (i-1)\Delta + 1$ and $q_i = i\Delta$, and then $\mathcal{I}_i = [p_i, q_i]$ is exactly the $i$-th stationary piece of the comparator sequence.

For any piece $i \in \{1, \ldots, \lceil T/\Delta\rceil\}$, we set all $u_t, t \in \mathcal{I}_i$ to be the best fixed decision $x_{\mathcal{I}_i}^\star$ regarding the cumulative linearized losses during interval $\mathcal{I}_i$, i.e., $u_t \equiv x_{\mathcal{I}_i}^\star = \arg\min_{x \in \mathcal{X}} \sum_{t \in \mathcal{I}_i} \lambda_t^\top F_t(x)$, for any $t \in \mathcal{I}_i, i \in \{1, \ldots, \lceil T/\Delta\rceil\}$. Then we can apply Lemma 4 to such comparator sequence and bound the term $A$ as

$$
\begin{aligned}
A &\leq \frac{\gamma}{\eta} \sum_{i=1}^{\lceil T/\Delta\rceil-1} \|u_{\mathcal{I}_i} - u_{\mathcal{I}_{i+1}}\| + \frac{1}{\eta} B_R(u_{\mathcal{I}_1}, x_1) + \frac{\eta}{2} \sum_{t=1}^{T} \|\nabla F_t(x_t) \lambda_t\|_*^2 \\
&\leq \frac{\gamma}{\eta} (\lceil \tfrac{T}{\Delta}\rceil - 1)D + \frac{\gamma D}{\eta} + \frac{\eta}{2} \sum_{t=1}^{T} \|\nabla F_t(x_t) \lambda_t\|_*^2 = \frac{\gamma D}{\eta} \lceil \tfrac{T}{\Delta}\rceil + \frac{\eta}{2} \sum_{t=1}^{T} \|\nabla F_t(x_t) \lambda_t\|_*^2.
\end{aligned}
$$

Notice that, the second term $B$ measures the difference between the cumulative linearized loss of the best decisions $x_{\mathcal{I}_i}^\star$ regarding each interval $\mathcal{I}_i$ and that of the comparators $\{x_t^*\}$. To analyze this term, we consider such difference $B_i$ restricted to any interval $\mathcal{I}_i, i \in \{1, \ldots, \lceil T/\Delta\rceil\}$:

$$
\begin{aligned}
B_i &= \sum_{t \in \mathcal{I}_i} \lambda_t^\top F_t(u_t) - \sum_{t \in \mathcal{I}_i} \lambda_t^\top F_t(x_t^*) \\
&= \sum_{t \in \mathcal{I}_i} \lambda_t^\top F_t(u_t) - \sum_{t \in \mathcal{I}_i} \lambda_{p_i}^\top F_{p_i}(x_{p_i}^*) + \sum_{t \in \mathcal{I}_i} \lambda_{p_i}^\top F_{p_i}(x_{p_i}^*) - \sum_{t \in \mathcal{I}_i} \lambda_t^\top F_t(x_t^*).
\end{aligned}
$$

Recall our definition of $u_t$ and $x_t^*$, then we have $\sum_{t \in \mathcal{I}_i} \lambda_t^\top F_t(u_t) \leq \sum_{t \in \mathcal{I}_i} \lambda_t^\top F_t(x_{p_i}^*)$ and $\lambda_{p_i}^\top F_{p_i}(x_{p_i}^*) \leq \lambda_{p_i}^\top F_{p_i}(x_t^*)$. Hence we further have

$$
B_i \leq \sum_{t \in \mathcal{I}_i} \lambda_t^\top F_t(x_{p_i}^*) - \sum_{t \in \mathcal{I}_i} \lambda_{p_i}^\top F_{p_i}(x_{p_i}^*) + \sum_{t \in \mathcal{I}_i} \lambda_{p_i}^\top F_{p_i}(x_t^*) - \sum_{t \in \mathcal{I}_i} \lambda_t^\top F_t(x_t^*).
$$

Moreover, for any $t \in \mathcal{I}_i, x \in \mathcal{X}$, we have

$$|\lambda_t^\top F_t(x) - \lambda_{p_i}^\top F_{p_i}(x)| = |\sum_{k=p_i}^{t-1}(\lambda_{k+1}^\top F_{k+1}(x) - \lambda_k^\top F_k(x))| \le \sum_{k=p_i}^{q_i} \sup_{x \in \mathcal{X}} |\lambda_k^\top F_k(x) - \lambda_{k+1}^\top F_{k+1}(x)|.$$

Recall that $\mathcal{I}_i$ has at most $\Delta$ elements, we further have

$$B_i \le 2\Delta \sum_{k=p_i}^{q_i} \sup_{x \in \mathcal{X}} |\lambda_k^\top F_k(x) - \lambda_{k+1}^\top F_{k+1}(x)|.$$

Hence the term $B$ can be bounded as

$$B = \sum_{i=1}^{\lceil T/\Delta \rceil} B_i \le 2\Delta \sum_{i=1}^{\lceil T/\Delta \rceil} \sum_{k=p_i}^{q_i} \sup_{x \in \mathcal{X}} |\lambda_k^\top F_k(x) - \lambda_{k+1}^\top F_{k+1}(x)|.$$

From the definition of $p_i$ and $q_i$, we further have

$$B \le 2\Delta \sum_{t=1}^{T-1} \sup_x |\lambda_t^\top F_t(x) - \lambda_{t+1}^\top F_{t+1}(x)|$$

$$\le 2\Delta \sum_{t=1}^{T-1} \sup_x (|\lambda_t^\top (F_t(x) - F_{t+1}(x))| + |(\lambda_t - \lambda_{t+1})^\top F_{t+1}(x)|).$$

Combining the above two inequalities on terms $A$ and $B$, we finally derive

$$R_{\text{MOD}}(T) \le \frac{\gamma}{\eta} \sum_{i=1}^{\lceil T/\Delta \rceil - 1} \|u_{\mathcal{I}_i} - u_{\mathcal{I}_{i+1}}\| + \frac{1}{\eta} B_R(u_{\mathcal{I}_1}, x_1) + \frac{\eta}{2} \sum_{t=1}^T \|\nabla F_t(x_t)\lambda_t\|_*^2$$

$$+ 2\Delta \sum_{t=1}^{T-1} \sup_{x \in \mathcal{X}} (|\lambda_t^\top (F_t(x) - F_{t+1}(x))| + |(\lambda_t - \lambda_{t+1})^\top F_{t+1}(x)|).$$

Assume that the diameter of the decision domain $\mathcal{X}$ is upper bounded by some $D > 0$, then $\|u_{\mathcal{I}_i} - u_{\mathcal{I}_{i+1}}\| \le D$ and $B_R(u_{\mathcal{I}_1}, x_1) \le \gamma \|u_{\mathcal{I}_1} - x_1\| \le \gamma D$. Hence we have

$$\frac{\gamma}{\eta} \sum_{i=1}^{\lceil T/\Delta \rceil - 1} \|u_{\mathcal{I}_i} - u_{\mathcal{I}_{i+1}}\| + \frac{1}{\eta} B_R(u_{\mathcal{I}_1}, x_1) \le \frac{\gamma D}{\eta}(\lceil T/\Delta \rceil - 1) + \frac{\gamma D}{\eta} = \frac{\gamma D}{\eta} \lceil \frac{T}{\Delta} \rceil.$$

Further assume that each loss function $F_t$ is coordinate-wise $G$-Lipschitz continuous, i.e., $|f_t^i(y) - f_t^i(x)| \le G\|y - x\|, \forall x, y \in \mathcal{X}, i \in \{1, \ldots, m\}$. Without loss of generality, we suppose $0 \in \mathcal{X}$ and $f_t^i(0) = 0, \forall t \in \{1, \ldots, T\}, i \in \{1, \ldots, m\}$. Then for any $x \in \mathcal{X}$, $|f_t^i(x)| \le GD$, and consequently

$$\sum_{t=1}^{T-1} |(\lambda_t - \lambda_{t+1})^\top F_{t+1}(x)| \le \sum_{t=1}^{T-1} \|\lambda_t - \lambda_{t+1}\|_1 \|F_{t+1}(x)\|_\infty \le 2F \sum_{t=1}^{T-1} \|\lambda_t - \lambda_{t+1}\|_1.$$

It then remains to tackle the term $\sum_{t=1}^{T-1} \sup_{x \in \mathcal{X}} |\lambda_t^\top (F_t(x) - F_{t+1}(x))|$. To this end, we introduce the following lemma.

**Lemma 5.** *For OMMD with composition weights $\lambda_t \in \Delta_m$ at each round $t$, it holds that*

$$\sum_{t=1}^{T-1} \sup_{x \in \mathcal{X}} |\lambda_t^\top (F_t(x) - F_{t+1}(x))| \le 2F(T - 1) \sum_{t=1}^{T-1} \|\lambda_t - \lambda_{t+1}\|_1 + V_T.$$

*Proof.* Denote $\overline{\lambda} = \frac{\sum_{k=1}^{T} \lambda_k}{T}$. We then decompose $\sum_{t=1}^{T-1} \sup_{x \in \mathcal{X}} |\lambda_t^{\top}(F_t(x) - F_{t+1}(x))|$ as

$$\sum_{t=1}^{T-1} \sup_{x \in \mathcal{X}} |\lambda_t^{\top}(F_t(x) - F_{t+1}(x))|$$

$$= \sum_{t=1}^{T-1} \sup_{x \in \mathcal{X}} |(\lambda_t - \overline{\lambda})^{\top}(F_t(x) - F_{t+1}(x)) + \overline{\lambda}^{\top}(F_t(x) - F_{t+1}(x))|$$

$$\leq \underbrace{\sum_{t=1}^{T-1} \sup_{x \in \mathcal{X}} |(\lambda_t - \overline{\lambda})^{\top}(F_t(x) - F_{t+1}(x))|}_{\text{term A}} + \underbrace{\sum_{t=1}^{T-1} \sup_{x \in \mathcal{X}} |\overline{\lambda}^{\top}(F_t(x) - F_{t+1}(x))|}_{\text{term B}}.$$

Since we have assumed that $|f_t^i(x)| \leq F$ for any $x \in \mathcal{X}, t \in \{1, \ldots, T\}, i \in \{1, \ldots, m\}$, it holds that $|f_t^i(x) - f_{t+1}^i(x)| \leq 2F$. Consequently, we can bound the term A as

$$\text{term A} = \sum_{t=1}^{T-1} \sup_{x \in \mathcal{X}} |\sum_{i=1}^{m} (\lambda_t^i - (\overline{\lambda})^i)(f_t^i(x) - f_{t+1}^i(x))|$$

$$\leq \sum_{t=1}^{T-1} \sup_{x \in \mathcal{X}} \sum_{i=1}^{m} |\lambda_t^i - (\overline{\lambda})^i| \cdot |f_t^i(x) - f_{t+1}^i(x)|$$

$$\leq \sum_{t=1}^{T-1} \sum_{i=1}^{m} |\lambda_t^i - (\overline{\lambda})^i| \cdot \sup_{x \in \mathcal{X}} |f_t^i(x) - f_{t+1}^i(x)|$$

$$\leq \sum_{t=1}^{T-1} \sum_{i=1}^{m} |\lambda_t^i - (\overline{\lambda})^i| \cdot 2F = \sum_{t=1}^{T-1} 2F \|\lambda_t - \overline{\lambda}\|_1,$$

where $\lambda_t^i$ and $(\overline{\lambda})^i$ represents the $i$-th entry of $\lambda_t$ and $\overline{\lambda}$, respectively. Also notice that for any $t \in \{1, \ldots, T\}$,

$$\|\lambda_t - \overline{\lambda}\|_1 = \|\frac{\sum_{k=1}^{T} \lambda_k}{T} - \lambda_t\|_1 = \|\frac{\sum_{k=1}^{T}(\lambda_k - \lambda_t)}{T}\|_1$$

$$= \|\frac{\sum_{k=1}^{T} \sum_{s=k}^{t-1}(\lambda_s - \lambda_{s+1})}{T}\|_1 \leq \sum_{k=1}^{T} \sum_{s=k}^{t-1} \|\frac{\lambda_s - \lambda_{s+1}}{T}\|_1$$

$$\leq \sum_{k=1}^{T} \sum_{s=1}^{T-1} \|\frac{\lambda_s - \lambda_{s+1}}{T}\|_1 = T \sum_{s=1}^{T-1} \|\frac{\lambda_s - \lambda_{s+1}}{T}\|_1 = \sum_{s=1}^{T-1} \|\lambda_s - \lambda_{s+1}\|_1.$$

Therefore, we have

$$\text{term A} \leq \sum_{t=1}^{T-1} 2F \sum_{s=1}^{T-1} \|\lambda_s - \lambda_{s+1}\|_1 = 2F(T-1) \sum_{t=1}^{T-1} \|\lambda_t - \lambda_{t+1}\|_1.$$

As for term B, we have

$$\text{term B} = \sum_{t=1}^{T-1} \sup_{x \in \mathcal{X}} |\overline{\lambda}^{\top}(F_t(x) - F_{t+1}(x))| \leq \sum_{t=1}^{T-1} \sup_{x \in \mathcal{X}} \sum_{i=1}^{m} (\overline{\lambda})^i \cdot |f_t^i(x) - f_{t+1}^i(x)|$$

$$\leq \sum_{i=1}^{m} (\overline{\lambda})^i \cdot \sum_{t=1}^{T-1} \sup_{x \in \mathcal{X}} |f_t^i(x) - f_{t+1}^i(x)| \leq \sum_{i=1}^{m} (\overline{\lambda})^i \cdot V_T = V_T,$$

where the last inequality is derived from the assumption of temporal variability, and the last equation comes from the fact that $\sum_{i=1}^{m} (\overline{\lambda})^i = \frac{1}{T} \sum_{i=1}^{m} \sum_{s=1}^{T} \lambda_s^i = \frac{1}{T} \sum_{s=1}^{T} \sum_{i=1}^{m} \lambda_t^i = 1$. Combining the above bounds for term A and term B, we finally prove the lemma. ∎

Plugging the above terms into the above bound for $R_{\text{MOD}}(T)$, and replace the quantity $\Delta \in \{1, \ldots, T\}$ by $\delta$, we have

$$R_{\text{MOD}}(T) \leq 2\delta V_T + 4\delta FT \sum_{t=1}^{T-1} \|\lambda_t - \lambda_{t+1}\|_1 + \frac{\eta}{2} \sum_{t=1}^{T} \|\nabla F_t(x_t)\lambda_t\|_*^2 + \frac{\gamma D}{\eta} \lceil \frac{T}{\delta} \rceil,$$

for any $\delta \in \{1, \ldots, T\}$. Note that $\sum_{t=1}^{T} \sup_{x \in \mathcal{X}}(|\lambda_t^\top(F_t(x) - F_{t+1}(x))| + |(\lambda_t - \lambda_{t+1})^\top F_{t+1}(x)|) = \sum_{t=1}^{T-1} \sup_{x \in \mathcal{X}}(|\lambda_t^\top(F_t(x) - F_{t+1}(x))| + |(\lambda_t - \lambda_{t+1})^\top F_{t+1}(x)|)$, because the regret does not depend on $F_{T+1}, \lambda_{T+1}$. We thus prove the lemma. ∎

### C.3  PROOF OF THEOREM 1

*Proof.* This theorem can be directly derived from Lemma 1.

Specifically, when $\frac{4V_T}{G^2T} \leq \eta \leq \frac{4V_T}{G^2}$, we can set $\delta = \frac{\eta G^2 T}{V_T}$, which satisfies $1 \leq \delta \leq T$. Plugging it into Lemma 1 and rearranging the inequality, we can directly derive the theorem. ∎

### C.4  PROOF OF THEOREM 2

*Proof.* Since this theorem is a special case of its following Theorem 3 when $\Omega(1) \leq V_T \leq o(T)$, it can be proved as we derive Theorem 3 in the following subsection.

In fact, as we assume $\Omega(1) \leq V_T \leq o(T)$, in the following derivation of Theorem 3, we are always in the case of (i). In addition, when $\Omega(1) \leq V_T \leq o(T)$ in Theorem 3 we exactly have $\eta = \frac{2}{G}(\frac{\gamma D V_T}{GT})^{1/3}$. Hence this theorem can be directly derived from Theorem 3. ∎

### C.5  PROOF OF THEOREM 3

*Proof.* Denote $\eta_0 = \frac{2}{G}(\frac{\gamma D V_T}{GT})^{1/3}$. We consider the following three cases:

(i) When $\frac{4V_T}{G^2T} \leq \eta_0 \leq \frac{4V_T}{G^2}$, we can directly apply the above lemma.

(ii) When $\eta_0 < \frac{4V_T}{G^2T}$, or equivalently $V_T > (\frac{\gamma D}{8})^{1/2}GT$, we have $\eta = \frac{4V_T}{G^2T}$. Set $\delta = 1$ in Lemma 1, then it can be verified that

$$R_{\text{MOD}}(T) \leq 2V_T + \frac{2V_T}{G^2T} \sum_{t=1}^{T} (\alpha \|\lambda_t - \lambda_{t-1}\|_1 + \|\nabla F_t(x_t)\lambda_t\|_*^2) + \frac{\gamma D G^2 T^2}{4V_T}$$

$$\leq 4V_T + \frac{2V_T}{G^2T} \sum_{t=1}^{T} \|\nabla F_t(x_t)\lambda_{t-1}\|_*^2$$

$$\leq 4V_T + \frac{2V_T}{G^2T} \sum_{t=1}^{T} G^2 = 6V_T.$$

(iii) When $\eta_0 > \frac{4V_T}{G^2}$, or equivalently $V_T < (\frac{\gamma D}{8T})^{1/2}G$, we have $\eta = \frac{4V_T}{G^2}$. Set $\delta = T$ in Lemma 1, then it can be verified that

$$R^{PT}(T) \leq 2TV_T + \frac{2V_T}{G^2} \sum_{t=1}^{T} (\alpha \|\lambda_t - \lambda_{t-1}\|_1 + \|\nabla F_t(x_t)\lambda_t\|_*^2) + \frac{\gamma D G^2}{4V_T}$$

$$\leq G(2\gamma D)^{1/2}T^{1/2} + \frac{2V_T}{G^2} \sum_{t=1}^{T} \|\nabla F_t(x_t)\lambda_{t-1}\|_*^2$$

$$\leq G(2\gamma D)^{1/2}T^{1/2} + \frac{2V_T}{G^2} \sum_{t=1}^{T} G^2 \leq \frac{3G}{2}(2\gamma D)^{1/2}T^{1/2}.$$

Combining (i)-(iii), we prove the theorem. ∎

## C.6 PROOF OF COROLLARY 1

*Proof.* Since this corollary is a special case of its following Corollary 2 when $\Omega(1) \leq V_T \leq o(T)$, it can be directly derived from Corollary 2.

Specifically, since $V_T \geq \Omega(1)$, we have $V_T^{1/3}T^{2/3} \geq O(T^{1/2})$. Moreover, since $V_T \leq o(T)$, we have $V^{1/3}T^{2/3} \geq o(V_T)$. Therefore, the dominating term in the bound of Corollary 2 is $V^{1/3}T^{2/3}$, which proves the corollary. ∎

## C.7 PROOF OF COROLLARY 2

*Proof.* We start from the general bound derived in Theorem 3. Specifically, in the first regret term, since $\lambda_t$ is selected to minimize $\alpha\|\lambda_t - \lambda_{t-1}\|_1 + \|\nabla F_t(x_t)\lambda_t\|_2^*$ at each step $t$, we further have

$$\min_{\lambda \in \Delta_m} \{\alpha\|\lambda - \lambda_{t-1}\|_1 + \|\nabla F_t(x_t)\lambda\|_*^2\} \leq \|\nabla F_t(x_t)\lambda_{t-1}\|_*^2 \leq (\|\lambda_{t-1}\|_1\|\nabla F_t(x_t)\|_\infty)^2 \leq G^2.$$

Plugging it into the bound in Theorem 3 directly proves the corollary. ∎

# D MORE DETAILS IN MULTI-OBJECTIVE STATIC REGRET

As we have discussed before, the multi-objective static regret cannot be formulated using most existing discrepancy metrics, since these metrics always give non-negative measurements, and the static regret based on any non-negative metric will fail to reduce to the standard static regret $R_S(T)$ in the single-objective setting. In this section, we give a possible form of multi-objective static regret and provide an analysis for it.

The static regret is enlightened by the equivalent form of dynamic regret derived in Proposition 1. Recall that in Proposition 1, the comparator $x_t^*$ at each round $t$ is selected from the Pareto set $\mathcal{X}_t^*$ of the instantaneous loss $F_t$; in addition, the weights $\lambda_t^*$ at each round $t$ is generated separately. To derive a static version, we can use a fixed comparator $x^*$ from the Pareto set $\mathcal{X}^*$ of the cumulative loss $\sum_t F_t$ and fixed weights $\lambda^*$ at all rounds. Now the static regret takes

$$R_{\text{MOS}}(T) := \sup_{x^* \in \mathcal{X}^*} \inf_{\lambda^* \in \Delta_m} \lambda^{*\top}(\sum_{t=1}^{T} F_t(x_t) - \sum_{t=1}^{T} F_t(x^*)),$$

where $\mathcal{X}^* = \mathcal{P}_X(\sum_{t=1}^{T} F_t)$. Note that when $m = 1$, the probabilistic simplex $\Delta_m$ reduces to a single point $\{1\}$, and the Pareto optimal set $\mathcal{X}^*$ coincides with the optimal set of the cumulative scalar loss $\sum_{t=1}^{T} F_t$, i.e., $\mathcal{X}^* = \arg\min_{x \in \mathcal{X}} F_t(x)$. Hence the multi-objective static regret takes $R_{\text{MOS}}(T) = \sup_{x \in \mathcal{X}^*}(\sum_{t=1}^{T} F_t(x_t) - \sum_{t=1}^{T} F_t(x^*)) = \sum_{t=1}^{T} F_t(x_t) - \min_{x \in \mathcal{X}} \sum_{t=1}^{T} F_t(x)$, which exactly reduces to the standard static regret $R_S(T)$ in the single-objective setting.

Surprisingly, with proper choices of $\eta$ and $\alpha$, our proposed OMMD-II algorithm can still achieve a sublinear bound w.r.t. $T$ for the static regret, as shown in the following theorem. Note that the derived regret bound for OMMD-II is tight w.r.t. $T$, since it matches the lower bound $O(\sqrt{T})$ of the standard static regret in the single-objective setting (Hazan, 2019).

**Theorem 4.** *OMMD-II with composition weights $\lambda_t$ attains the following static regret*

$$R_{\text{MOS}}(T) \leq 2FT \sum_{t=1}^{T-1} \|\lambda_t - \lambda_{t+1}\|_1 + \frac{1}{\eta}B_R(x^*, x_1) + \frac{\eta}{2}\sum_{t=1}^{T} \|\nabla F_t(x_t)\lambda_t\|_*^2. \tag{7}$$

*By setting $\eta = \frac{\sqrt{2D}}{G\sqrt{T}}$ and $\alpha = \frac{4FT}{\eta}$, the regret bound reduces to $O(\sqrt{T})$, which is sublinear w.r.t. $T$.*

*Proof.* We start from the definition of $R_{\text{MOS}}(T)$. Specifically, for any $\lambda \in \Delta_m$ and $\lambda_1 \ldots, \lambda_T \in \Delta_m$, it holds that

$$R_{\text{MOS}}(T) = \sup_{x^* \in \mathcal{X}^*} \inf_{\lambda^* \in \Delta_m} \sum_{t=1}^{T} \lambda^{*\top}(F_t(x_t) - F_t(x^*)) \leq \sup_{x^* \in \mathcal{X}^*} \sum_{t=1}^{T} \lambda^{\top}(F_t(x_t) - F_t(x^*))$$

$$= \sum_{t=1}^{T} ((\lambda^{\top} F_t(x_t) - \lambda_t^{\top} F_t(x_t)) + \lambda_t^{\top}(F_t(x_t) - F_t(x^*)) + (\lambda_t^{\top} F_t(x^*) - \lambda^{\top} F_t(x^*)))$$

$$\leq \sum_{t=1}^{T} F\|\lambda - \lambda_t\|_1 + \sum_{t=1}^{T} \lambda_t^{\top}(F_t(x_t) - F_t(x^*)) + \sum_{t=1}^{T} F\|\lambda - \lambda_t\|_1$$

$$= 2F \sum_{t=1}^{T} \|\lambda - \lambda_t\|_1 + \sum_{t=1}^{T} \lambda_t^{\top}(F_t(x_t) - F_t(x^*)).$$

To tackle the second term in the above inequality, we set $u_1 = u_2 = \ldots = u_T = x^*$ in Lemma 4, which results in

$$\sum_{t=1}^{T} \lambda_t^{\top}(F_t(x_t) - F_t(x^*)) \leq \frac{1}{\eta} B_R(x^*, x_1) + \frac{\eta}{2} \sum_{t=1}^{T} \|\nabla F_t(x_t)\lambda_t\|_*^2.$$

To analyze the static regret for OMMD-II, we relate the first term $\sum_{t=1}^{T} \|\lambda - \lambda_t\|_1$ with the $L1$ regularization in the regularized min-norm solver at each round. Specifically, we denote the average weights as $\bar{\lambda} = \sum_{t=1}^{T} \lambda_t / T$ and set $\lambda = \bar{\lambda}$ in the first term. Then for any $t \in \{1, \ldots, T\}$, we have

$$\|\bar{\lambda} - \lambda_t\|_1 = \|\frac{\sum_{i=1}^{T}(\lambda_i - \lambda_t)}{T}\|_1 = \|\frac{\sum_{i=1}^{T} \sum_{k=i}^{t-1}(\lambda_k - \lambda_{k+1})}{T}\|_1$$

$$\leq \sum_{i=1}^{T} \sum_{k=i}^{t-1} \|\frac{\lambda_k - \lambda_{k+1}}{T}\|_1 \leq \sum_{i=1}^{T} \sum_{k=1}^{T-1} \|\frac{\lambda_k - \lambda_{k+1}}{T}\|_1$$

$$= T \sum_{k=1}^{T-1} \|\frac{\lambda_k - \lambda_{k+1}}{T}\|_1 = \sum_{t=1}^{T-1} \|\lambda_t - \lambda_{t+1}\|_1.$$

Consequently, the static regret can be bounded as

$$R_{\text{MOS}}(T) \leq 2F \sum_{t=1}^{T} \sum_{t=1}^{T-1} \|\lambda_t - \lambda_{t+1}\|_1 + \frac{1}{\eta} B_R(x^*, x_1) + \frac{\eta}{2} \sum_{t=1}^{T} \|\nabla F_t(x_t)\lambda_t\|_*^2$$

$$= 2FT \sum_{t=1}^{T-1} \|\lambda_t - \lambda_{t+1}\|_1 + \frac{1}{\eta} B_R(x^*, x_1) + \frac{\eta}{2} \sum_{t=1}^{T} \|\nabla F_t(x_t)\lambda_t\|_*^2,$$

which proves the full static regret bound. Now we prove the reduced bound. From the full bound, we equivalently have

$$R_{\text{MOS}}(T) \leq \frac{1}{\eta} B_R(x^*, x_1) + \frac{\eta}{2} \sum_{t=1}^{T} (\|\nabla F_t(x_t)\lambda_t\|_*^2 + \frac{4FT}{\eta}\|\lambda_t - \lambda_{t-1}\|_1).$$

We now set $\lambda = \frac{4FT}{\eta}$ in OMMD-II, and specify $\lambda_t$ to be the composition weights generated by the algorithm at round $t$. Then we know that $\lambda_t \in \arg\min_{\lambda \in \Delta_t} \|\nabla F_t(x_t)\lambda\|_*^2 + \frac{4FT}{\eta}\|\lambda - \lambda_{t-1}\|_1$. In particular, we have $\|\nabla F_t(x_t)\lambda_t\|_*^2 + \frac{4FT}{\eta}\|\lambda_t - \lambda_{t-1}\|_1 \leq \|\nabla F_t(x_t)\lambda_{t-1}\|_*^2$. Therefore, it holds that

$$R_{\text{MOS}}(T) \leq \frac{1}{\eta} B_R(x^*, x_1) + \frac{\eta}{2} \sum_{t=1}^{T} \|\nabla F_t(x_t)\lambda_{t-1}\|_*^2.$$

Utilize Assumption 1 and Assumption 3, and set $\eta = \frac{\sqrt{2D}}{G\sqrt{T}}$, then we have

$$R_{\text{MOS}}(T) \leq \frac{1}{\eta} B_R(x^*, x_1) + \frac{\eta}{2} \sum_{t=1}^{T} \|\nabla F_t(x_t)\lambda_{t-1}\|_*^2 \leq G\sqrt{2DT},$$

which proves the reduced bound. ■

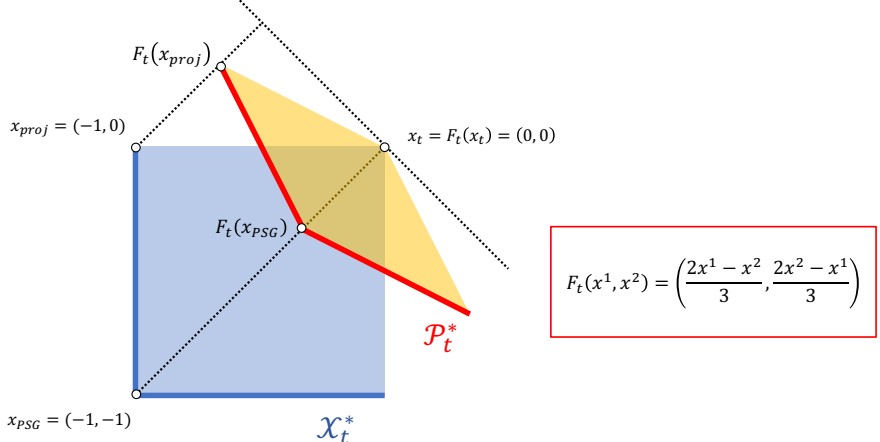

Figure 4: Illustration of an example in which the projection-based metric $\Delta_t^{proj}$ fails to measure the Pareto optimality of the generated decision $x_t$. The light blue area and the orange area represent the decision set $\mathcal{X}$ and its image set $F_t(\mathcal{X})$, respectively. The blue line segments constitute the Pareto optimal set $\mathcal{X}_t^*$. The red line segments constitute the Pareto front $\mathcal{P}_t^*$. As shown in the plot, in this example, the projection-based metric $\Delta_t^{proj}$ compares $F_t(x_t)$ with the farthest point $F_t(x_{proj})$ in the Pareto front; the comparator $x_{proj}$ does not even dominate $x_t$. In contrast, PSG compares $F_t(x_t)$ with the nearest point $F_t(x_{PSG})$ in the Pareto front; the comparator $x_{PSG}$ indeed dominates $x_t$.

Note that the newly introduced static regret is no longer based on PSG. It can be understood as induced from a new discrepancy metric $\delta(x_t; x^*, F_t, \lambda^*) = (\lambda^*)^\top (F_t(x_t) - F_t(x^*))$, where $\lambda^* \in \Delta_m$. We thus have $R_{\text{MOS}}(T) = \sup_{x^* \in \mathcal{X}^*} \inf_{\lambda^* \in \Delta_m} \sum_{t=1}^T \delta(x_t; x^*, F_t, \lambda^*)$. Such a metric compares the generated decision $x_t$ at each round with the fixed comparator $x^* \in \mathcal{X}^*$. Hence, in particular, it is able to produce a negative value when $x_t$ dominates $x^*$ regarding the instantaneous loss $F_t$, making it a general extension of PSG. Although this property is desired in the definition of static regret, it is rarely utilized in the discrepancy metrics in multi-objective optimization. Thus it looks a bit strange as a Pareto suboptimality metric and hence its physical meaning needs to be justified further. Besides, there are undoubtedly many other possible ways to define such metrics. Therefore, regarding the multi-objective static regret, much work remains to be done in the future. We hope our initial attempt paves the way for future research.

## E DISCUSSION ON AN ALTERNATIVE METRIC BASED ON PROJECTION

In this section, we discuss an alternative discrepancy metric based on projection onto the Pareto optimal set. Then we explain why it is unsuitable to measure the Pareto suboptimality in some cases.

The metric is formulated as follows, which we term $\Delta_t^{proj}$. At each round $t$, we project the generated decision $x_t$ onto the Pareto optimal set $\mathcal{X}_t^*$, namely $x_{proj} \in \arg\min_{x' \in \mathcal{X}_t^*} \|x_t - x'\|_2$, and then measure the Euclidean distance between $F_t(x_t)$ and $F_t(x_{proj})$, i.e., $\Delta_t^{proj}(x_t) = \|F_t(x_t) - F_t(x_{proj})\|_2$. Different from the PSG metric $\Delta_t$ that directly measures the distance between the actual loss $F_t(x_t)$ and the entire Pareto front $\mathcal{P}_t^*$ in the loss space, the projection-based metric $\Delta_t^{proj}$ only compares the actual loss $F_t(x_t)$ with the loss $F_t(x_{proj})$ evaluated at a single comparator $x_{proj}$.

In intuition, $\Delta_t^{proj}$ ignores the landscape of the Pareto front, since the choice of $x_{proj}$ only depends on the Pareto set, whose structure in the decision domain does not necessarily align with the landscape of the Pareto front in the loss space. Therefore, the comparator $x_{proj}$ may not be a good point to measure the Pareto suboptimality of $x_t$ in the loss space. In comparison, PSG directly compares

$F_t(x_t)$ with the entire Pareto front, thus is always able to measure the Pareto suboptimality of the generated decision $x_t$.

In the following, we provide an example in the two-objective convex setting to corroborate this point. In our example, the loss $F_t(x_{proj})$ induced by the projected decision $x_{proj}$ is actually the most remote point from $F_t(x_t)$ in the Pareto front $\mathcal{P}_t^*$. Moreover, for the second objective, we have $f_t^2(x_{proj}) > f_t^2(x_t)$. Therefore, $x_{proj}$ is not a proper point to measure the Pareto suboptimality of $x_t$ regarding $\mathcal{P}_t^*$. Notably, measuring the Euclidean distance between $F_t(x_{proj})$ and $F_t(x_t)$ is meaningless, since $x_{proj}$ performs even worse than $x_t$ in the second objective.

Concretely, we consider the decision set $\mathcal{X} = \{(x^1, x^2) \mid -1 \leq x^1 \leq 0, -1 \leq x^2 \leq 0\}$, which is a rectangle in $\mathbb{R}^2$. We assume there are two objectives, and at round $t$, the loss function $F_t : \mathcal{X} \to \mathbb{R}^2$ takes $F_t(x^1, x^2) = (\frac{2x^1 - x^2}{3}, \frac{2x^2 - x^1}{3})$. Since $F_t$ is a linear transformation, it is convex. The decision set $\mathcal{X}$ and the image set $F_t(\mathcal{X})$ are shown in Figure. It is easy to verify that, the Pareto optimal set $\mathcal{X}_t^* = \{(-1, x^2) \mid -1 \leq x^2 \leq 0\} \cup \{(x^1, -1) \mid -1 \leq x^1 \leq 0\}$, which consists of two line segments. We assume the generated decision $x_t = (0, 0)$ at this round $t$, which incurs the loss $F_t(x_t) = (0, 0)$. Then the projection of $x_t$ on the Pareto set is $x_{proj} = (-1, 0)$ or $(0, -1)$. Due to symmetry, we only consider $x_{proj} = (-1, 0)$, then the loss evaluated at the comparator is $F_t(x_{proj}) = (-\frac{2}{3}, \frac{1}{3})$. From the plot in Figure, it is obvious that $F_t(x_{proj})$ is the most remote point from $F_t(x_t)$ in the Pareto front. Moreover, regarding the second objective, $f_t^2(x_{proj}) > f_t^2(x_t)$, which means $x_{proj}$ does not even dominate $x_t$.

As a comparison, we also investigate PSG in the above example. Recall that, in Proposition 1, PSG $\Delta_t(x_t)$ equals $\sup_{x \in \mathcal{X}_t^*} \inf_{\lambda \in \Delta_2} \lambda^\top (F_t(x_t) - F_t(x))$, or equivalently $\sup_{x \in \mathcal{X}_t^*} \min_{i \in \{1,2\}} (f_t^i(x_t) - f_t^i(x))$. It is easy to verify that $\Delta_t(x_t) = \frac{1}{3}$ and the comparator $x_t^*$ attaining the supremum is $(-1, -1)$, which we denote as $x_{PSG}$. Then the compared loss $F_t(x_{PSG}) = (-\frac{1}{3}, -\frac{1}{3})$, which is exactly the closest point to $F_t(x_t)$ in the Pareto front. Moreover, $x_{PSG}$ dominates $x_t$. Hence, compared to the projection-based metric $\Delta_t^{proj}$ that compares $F_t(x_t)$ with the most remote point in the Pareto front, PSG is obviously more reasonable to measure the Pareto suboptimality in the multi-objective setting.

