# OpenReview forum: "Multi-Objective Online Learning"
_ICLR.cc/2022/Conference — ICLR 2022 Submitted_

### Official Review · Reviewer_Qnvq · 2021-10-31

**Correctness:** 4
**Technical Novelty And Significance:** 3
**Empirical Novelty And Significance:** 3
**Recommendation:** 6
**Confidence:** 3

**Main Review:**

The paper addresses an important question to bound the regret of online multi-objective optimization. The derived bound of O(mT^{1/3}V_T^{2/3}) is also an interesting result that scales with the number of objectives m, time horizon T, and the regularity metric V_T. Here are my comments and suggestions:

1 - The regret bound in Theorem 1 is a main result of the paper. However, the theoretical techniques used in its proof seem to be fairly standard. A summary of the key and challenging analytical steps in the derivation can be helpful.

2 - A deep review of the problem background is given in sections 1 and 2. However, the focus here is to describe the problem and its importance. As a consequence, several closely related works on the dynamic regret of online mirror descent are missed in the literature review, namely:

- Ali Jadbabaie, Alexander Rakhlin, Shahin Shahrampour, and Karthik Sridharan. Online Optimization: Competing with Dynamic Comparators. Artificial Intelligence and Statistics (AISTATS), 2015.

- Shahin Shahrampour and Ali Jadbabaie, Distributed Online Optimization in Dynamic Environments Using Mirror Descent. IEEE Transactions on Automatic Control (TAC), March 2018

- Nima Eshraghi and Ben Liang. On the dynamic regret of Online Multiple Mirror Descent. Publicly available at https://openreview.net/forum?id=RepN5K31PT3

The last reference seems to have a similar idea and even acronym OMMD, but in over a single-objective online learning domain.

4 - Assumption 1 constrains the use of regularization function by imposing a Lipschitz condition. I wonder how often this assumption holds? Can the authors provide several examples and clarify the order of the parameter $\gamma$? Is this assumption satisfied in the examples used in the experiments?

5 - The dynamic regret described in section 3 compares the algorithm performance with a sequence of optimizers (single-objective case). However, there exists a more general form of dynamic regret introduced in the seminal work of (Zinkevich, 2003) which allows the use of arbitrary comparison targets.
The paper extends the dynamic regret definition to multi-objective case by considering a sequence of pareto optimal points. However, the general form of dynamic regret still can be used in multi-objective case. Is it possible to bound the general form of dynamic regret (comparison against arbitrary target sequence) using OMMD algorithm?

6 - The paper is mostly clear, and I enjoyed reading it. However, there are some minor unclear parts:
In Lemma/Theorem statements the algorithm is referred to as OMMD. However, two different updates are provided in the algorithm block labelled as OMMD-I and OMMD-II. I suggest making this style uniform for ease of reading.

**Summary Of The Paper:**

The paper studies the problem of multi-objective optimization in an online setting, where the objective functions are time-varying. An algorithm is proposed to solve this problem. The algorithm queries the gradient of all objective functions, and returns a convex combination of these gradients as the input to the mirror descent algorithm. The performance of the algorithm is measured compared to a sequence of Pareto optimal points, where a single component of the loss vector cannot be improved without causing harm to the other components. Under some set of assumptions, Theorem 1 provides an upper bound on the dynamic regret of the algorithm.

**Summary Of The Review:**

The paper addresses an important question to bound the regret of online multi-objective optimization. The derived bound of O(mT^{1/3}V_T^{2/3}) is also an interesting result that scales with the number of objectives m, time horizon T, and the regularity metric V_T. There are some concerns about the connection to the literature, assumptions, technical and analytical steps, which are all detailed above.

---

> ### Author Response · Authors · 2021-11-23
> **Author Response**
>
> Thank you for your helpful review!
>
> **Q1: “The theoretical techniques used in its proof seem to be fairly standard. A summary of the key and challenging analytical steps in the derivation can be helpful.”**
>
> **A1:** We highlight two technical novelties in the derivation of regret bounds, which are unique characteristics of analyzing the dynamic regret in the multi-objective setting.
>
> **(1)** In Proposition 1, we discover a novel connection between the dynamic regret and the zero-order PSG metric. Specifically, we derive an equivalent form of the dynamic regret by transforming PSG into a first-order form, making further regret analysis possible. The derivation is essential and highly non-trivial, as is well approved by Reviewer ieZJ.
>
> **(2)** In the proof of Lemma 1 (page 17 in the revision), we decompose the term $\|\lambda_t^\top F_t(x) - \lambda\_{t+1}^\top F\_{t+1}(x)\|$ at each round $t$ into a functional variation term $\|\lambda_t^\top(F_t(x) - F\_{t+1}(x))\|$ with fixed composition weights $\lambda_t$ and a weight variation term $\|(\lambda_t - \lambda\_{t+1})^\top F\_{t+1}(x)\|$ with a fixed function $F\_{t+1}$. Notably, for the former term, we utilize the stability of the composition weights $\lambda_t$ produced by our algorithm and relate this term to the temporal variability of each objective, which is novel and non-trivial. The detailed analysis is given in the proof of Lemma 5.
>
> **Q2: “several closed related works on the dynamic regret of online mirror descent are missed”**
>
> **A2:** Good suggestion! We have added your mentioned literature in the revision.
>
> **Q3: “Assumption 1 constraints the use of regularization function by imposing a Lipschitz condition. I wonder how often this assumption holds? Can the authors provide several examples and clarify the order of the parameter $\gamma$? Is this assumption satisfied in the experiments?”**
>
> **A3:** In fact, utilizing the Lipschitz condition is a common practice in the dynamic regret analysis of online mirror descent (for example, in [1] and your mentioned [2, 3]). If we adopt the L2 regularizer, the Bregman divergence is exactly the squared Euclidean distance $B_R(x,y)=\Vert x-y\Vert^2/2$, then OMD essentially reduces to online gradient descent. In this case, the Lipschitz parameter is bounded by the diameter $D$ of the decision set $\mathcal X$. In our experiments, we adopt the L2 regularizer and set the diameter of the decision set as $10$, hence the Lipschitz parameter $\gamma \leq10$.
>
> **Q4: “a more general form of dynamic regret … which allows the use of arbitrary comparison targets. Is it possible to bound the general form of dynamic regret (comparison against arbitrary target sequence) using the OMMD algorithm?”**
>
> **A4:** Admittedly, comparing against arbitrary target sequences with a bounded path length is another common way to analyze the dynamic regret. We have actually considered this approach before, but find that it is not straightforward to extend this form to the multi-objective setting. Specifically, recall that in all existing Pareto suboptimality metrics such as PSG and Hypervolume, the comparator is actually the whole Pareto set, NOT a single target decision. If we only use the single target sequence, the information on the Pareto set will be lost.
>
> Therefore, a more reasonable way to extend your suggested regret form is to use a sequence of target sets rather than a sequence of target decision points. However, how to define the path length of the set sequence remains an open question by itself. For example, we need to define the distance between two given decision sets.
>
> Given the above difficulties, we adopt the regret based on temporal variability, which is also very common for the dynamic regret analysis [4, 5].
>
> **Q5: “minor unclear parts … making this style uniform”**
>
> **A5:** Good suggestion! In the revision, we have replaced “OMMD” in the statements of Lemma 1 and Theorem 1 by “OMMD-I or OMMD-II” to make the style uniform.
>
> **Q6: “the derived bound of $O(m^{1/3}V_T^{1/3}T^{2/3})$ is an interesting result that scales with the number of objectives $m$”**
>
> **A6:** We have slightly modified the theoretical regret bound, which is now irrelevant of $m$. We also analyze the reason why $m$ is irrelevant and prove the tightness of the derived bound. Please refer to the **General Author Response (Part 1)** and the revision for more details.
>
> [1] A Simple Online Algorithm for Competing with Dynamic Comparators. Zhang et al. UAI 2020.
>
> [2] Distributed Online Optimization in Dynamic Environments Using Mirror Descent. Shahin Shahrampour and Ali Jadbabaie. TAC 2018.
>
> [3] Online Optimization: Competing with Dynamic Comparators. Jadbabaie et al. AISTATS 2015.
>
> [4] Non-stationary Stochastic Optimization. Besbes et al. Operations Research 2015.
>
> [5] Dynamic Regret of Strongly Adaptive Methods. Zhang et al. ICML 2018.

---

### Official Review · Reviewer_3rTx · 2021-11-01

**Correctness:** 3
**Technical Novelty And Significance:** 3
**Empirical Novelty And Significance:** Not applicable
**Recommendation:** 6
**Confidence:** 3

**Main Review:**

Strengths: A novel framework is proposed. Two new algorithms which both theoretically and experimentally work in the latter framework are designed.
Weaknesses:
(1) The suboptimality of $x_t$ in each round is measured using the Pareto suboptimality gap. A better measure of suboptimality could be the euclidean distance between the loss vector $F_t(x_t)$ and $F_t(x^*)$, where $x^*$ is the closest point to $x_t$ from the Pareto Optimal Decision set.
(2) The most common notion of regret is the static regret which is not bounded by the two proposed algorithms. Moreover, although I read the Remark at page 5, it is not clear to me why having a discrepancy metric which could be negative is a problem. In the classic online convex optimization setting it is possible to have $F_{t_0}(x_{t_0}) - F_{t_0}(x^*) < 0$ for a specific round $t_0$ since $x^*$ optimizes the overall sum $\sum_t F_t(x)$.

**Summary Of The Paper:**

The paper formulates a novel framework for multi-objective online convex optimization. The novel framework, similarly to the single-objective online convex optimization framework, can be viewed as a two players repeated game where at each round the online learner selects a point $x_t$ and the (possibly adversarial) environment selects a vector valued loss function $F_t (\cdot )$. To extend the notion of regret to the multi-objective setting at each round the suboptimality of $x_t$ is measured using the Pareto suboptimality gap (PSG). Two algorithms (OMMD-I and OMMD-II) which upper bound the extension of the dynamic regret in the multi-objective setting are designed. The two algorithms can be viewed as extensions of the min-norm method for offline multi-objective optimization in the online setting. In both algorithms a composite gradient, which is a convex combination of the descent directions for every objective, is calculated. This composite gradient is used to select, together with a Bregman reguralization, the descent direction and consequently the point $x_{t+1}$. The main difference between the two algorithms OMMD-I and OMMD-II is that in the latter the composition of the descent directions in two consecutive rounds will possibly be "more similar" (due to the regularization term $|| \lambda - \lambda_{t-1}||_1$).

**Summary Of The Review:**

The main weakness of the paper is that the static regret is not bounded by the proposed algorithms.

---

> ### Author Response · Authors · 2021-11-23
> **Author Response**
>
>  Thank you for your thoughtful review!
>
> **Q1: “A better measure of suboptimality could be the Euclidean distance between $F_t(x_t)$ and $F_t(x^\*)$, where $x^\*$ is the closest point to $x_t$ from the Pareto optimal set.”**
>
> **A1:** We have checked this metric carefully but find it unsuitable for measuring the Pareto suboptimality in the multi-objective setting. We explicate this in the following.
>
> As introduced in Section 2.2, the Pareto suboptimality is measured in the loss space; a desirable Pareto suboptimality metric should compare the actual loss $F_t(x_t)$ with the entire Pareto front, like PSG or Hypervolume. However, this projection-based metric ignores the landscape of the Pareto front, as the comparator $x^*$ used in it only depends on the Pareto set in the decision space. In general, the landscape of the Pareto front is usually quite different from that of the Pareto set. Therefore, the loss $F_t(x^*)$ at the comparator $x^*$ may not be an appropriate point to be used to measure the distance between $F_t(x_t)$ and the Pareto front. In other words, directly optimizing this metric may incur the situation that the loss of one objective is decreased while the loss of some other objective is largely increased.
>
> To better illustrate this point, we provide an example in the revision (Section E in the appendix), in which the projection-based metric $\Delta_t^{proj}$ actually compares $F_t(x_t)$ with the farthest point $F_t(x\_{proj})$ in the Pareto front, and the comparator $x_{proj}$ does not even dominate $x_t$. Hence $\Delta_t^{proj}$ will fail to measure the Pareto suboptimality of $x_t$ in this case. Specifically, we consider a two-objective optimization scenario and set the decision set as $\mathcal X=\\{(x^1,x^2)\mid -1\leq x^1\leq 0,-1\leq x^2\leq 0\\}\subset\mathbb R^2$. We further assume the generated decision $x_t=(0,0)$ and the convex loss function $F_t(x^1,x^2) = (\\frac{2x^1-x^2}{3}, \\frac{2x^2-x^1}{3})$ at some round $t$, then $F_t(x_t)=(0,0)$. In this case, it is easy to verify that the Pareto set of $F_t$ consists of two line segments: one lies between $(-1,-1)$ and $(-1,0)$, and the other lies between $(-1,-1)$ and $(0,-1)$. Therefore, the projection of $x_t=(0,0)$ onto the Pareto set is $(-1,0)$ or $(0,-1)$. For simplicity, we assume $x_{proj}=(-1,0)$, then $F_t(x_{proj})=(-\frac{2}{3},\frac{1}{3})$. It can then be verified that $F_t(x_{proj})$ is actually the farthest point from $F_t(x_t)$ in the Pareto front (see Figure 4 in the revision). Moreover, the comparator $x_{proj}$ does not even dominate $x_t$, since $f_t^2(x_{proj})=\frac{1}{3}>f_t^2(x_t)$ for the second objective. In fact, in this case, optimizing toward $x_{proj}$ will even increase the loss of the second objective. Therefore, $x_{proj}$ is not a proper comparator to measure the Pareto suboptimality of $x_t$ in this case.
>
> Note that in comparison, in the above example, PSG uses the comparator $x_{PSG}=(-1,-1)$, which incurs a loss $F_t(x_{PSG})=(-\frac{1}{3},-\frac{1}{3})$. Hence $x_{PSG}$ indeed dominates $x_t$. In addition, we can show that $F_t(x_{PSG})$ is the nearest point to $F_t(x_t)$ in the Pareto front (also see Figure 4 in the revision). Therefore, in this case, PSG indeed provides a measurement on the Pareto suboptimality of $x_t$ regarding $F_t$.
>
> **Q2: “the static regret is not bounded by the two proposed algorithms. In the Remark at page 5, it is not clear to me why having a discrepancy metric which could be negative is a problem.”**
>
> **A2:** We have addressed your concern on the static regret in the **General Author Response (Part 2)**. Admittedly, we can provide a possible form of the static regret in the multi-objective setting, which can also be optimized by our proposed OMMD-II algorithm. However, as discussed in the Remark at page 5, the formulation of the static regret requires a new discrepancy metric that can yield negative values when the generated decision $x\_{t_0}$ dominates the comparator $x^*$ regarding the instantaneous loss $F\_{t_0}$ for a specific round $t_0$. Although this property can be naturally satisfied by the scalar subtraction $F\_{t_0}(x\_{t_0})-F\_{t_0}(x^*)$ in the single-objective setting as you suggested, it is rarely utilized in the Pareto suboptimality metrics in multi-objective optimization. Hence the discrepancy metric induced from our provided static regret looks a bit strange as a Pareto suboptimality metric, which is very complicated to be understood. As the first research on multi-objective online convex optimization, we choose to analyze the dynamic regret, which we find closely related to the PSG metric that has been widely recognized in related fields. In fact, our newly introduced static regret and related analysis can be also included in the main body of the paper to make the discussion more comprehensive.

---

### Official Review · Reviewer_9N6u · 2021-11-02

**Correctness:** 4
**Technical Novelty And Significance:** 3
**Empirical Novelty And Significance:** 3
**Recommendation:** 6
**Confidence:** 3

**Main Review:**

As indicated above, the multi-objective online learning problem has already been studied in the bandit setting, but the adversarial full-information setting was left open. The model of Pareto regret defined in terms of the Pareto Suboptimality Gap is relevant, but was already introduced in (Drugan & Nowe, 2013; Turgay et al., 2018; Lu et al., 2020). So, the real technical novelty of this paper does not lie in the formalization of the problem, but in the derivation of new (Pareto) regret bounds in the full-information setting.

To this point, I was a bit confused by the results: using the online convex optimization framework, and more specifically the online mirror descent paradigm, the authors could derive a bound for the dynamic version of the regret, but not for the static version, which is actually a specific case (of the dynamic version). I understand that the key result, summarized in Proposition 1, only holds for the dynamic case, but this does not prevent us from finding a different approach to the static case.

For the static version, it seems that one can still use the standard version of OMD by pursuing a single objective. Indeed, we know that for each $i \in [m]$, the point $x^*_i$ is Pareto optimal. Furthermore, unless I missed something, the definition of PSG (Definition 3), implies that the instantaneous Pareto regret $\Delta_t$ is upper bounded by the instantaneous regret $f^i_t(x_t) - f^i_t(x^*_i)$ of any objective $i$, provided that each comparator set $\mathcal X^*_t$ coincides with the Pareto set induced from the cumulative loss $\sum_t F_t$. Under this assumption, and using the optimal bound in $O(\sqrt T)$ achieved by OMD for the single-objective case, we should have a bound in $O(\sqrt T)$ for the multi-objective case.

Nevertheless, for this dynamic version, the bound achieved by the online multi-objective mirror descent algorithm (second version) looks tight.  By the way, it would be nice to provide a lower bound here, in order to corroborate this tightness. Notably is $m^{1/3}$ tight?


**Summary Of The Paper:**

In the standard online learning model, the aim of the learning algorithm is to minimize its cumulative regret, defined by the difference in cumulative losses between the algorithm and the best decision taken with the benefit of insight in some reference set. In the generalized model of “multi-objective” online learning, the feedback is no longer a single scalar, but a vector of losses, each defined for a distinct objective. The performance of the learning algorithm is often defined in terms of “Pareto regret”, for which each instantaneous regret is captured by the notion of Pareto Suboptimality Gap (PSG). Although multi-objective online learning has been studied in the (possibly contextual and linear) bandit setting, many questions remain open. Notably, this paper focuses on the full-information setting where a vector-valued loss function is supplied at the end of each round. Based on the Online Mirror Descent (OMD) paradigm, two versions of multi-objective OMD are provided and analyzed for the dynamic version of Pareto regret (the static version being left as an open issue). Comparative experiments for both versions are performed on several benchmarks.


**Summary Of The Review:**

This paper provides a non-trivial regret bound for the multi-objective online learning problem in the full-information, yet possibly adversarial, setting, using the online convex optimization framework. As far as I know, this is a novel result. However, I am not entirely convinced that this is significant enough for the ICLR conference since this Pareto regret bound only holds in the dynamic setting. Furthermore, it would be interesting to provide a lower bound in order to justify the tightness of this bound.

*** After Rebuttal Phase ***

The authors have addressed several issues, notably related to the tightness of the bound for the dynamic regret, and an $O(\sqrt T)$ bound for the static regret. So, I have slightly changed my score. I would recommend revising the paper by incorporating the interesting results obtained in Section D.

---

> ### Author Response · Authors · 2021-11-23
> **Author Response**
>
> Thank you for your detailed review!
>
> **Q1: “the definition of PSG implies that the instantaneous Pareto regret $\Delta_t$ is upper bounded by the instantaneous regret $f^i_t(x_t)-f^i_t(x_t^\*)$ of any objective $i$, provided that each comparator set $\mathcal X^\*_t$ coincides with the Pareto set induced from the cumulative loss $\sum^T\_{t=1} F_t$. Using the optimal bound in $O(\sqrt T)$ achieved by OMD for the single-objective case, we should have a bound in $O(\sqrt T)$ for the multi-objective case.”**
>
> **A1:** In fact, in our derivation of Proposition 1, the instantaneous Pareto regret satisfies $\Delta_t\leq f^i_t(x_t)-f^i_t(x^*_t)$ for any $i\in\\{1,\dots,m\\}$ only when the generated decision $x_t$ is dominated by the comparator $x^*_t$ regarding the instantaneous loss $F_t$, i.e., $f^i_t(x_t)\geq f^i_t(x^*)$ for any $i\in\\{1,\dots,m\\}$. Note that this condition naturally holds for the dynamic regret where $x^*_t\in\mathcal X^*_t$. However, it does not necessarily holds for the static regret where $x^*_t$ is selected from the Pareto set $\mathcal X^*$ of the cumulative loss $\sum^T\_{t=1} F_t$. The reason is that, your mentioned assumption, i.e., $\mathcal X^*_t$ coincides with $\mathcal X^*$, **does NOT hold in the adversarial setting**. Specifically, in the adversarial setting, the instantaneous losses $F_t$ at different rounds may vary widely, hence $F_t$ may be very different from the cumulative loss $\sum^T\_{t=1} F_t$. As a consequence, the Pareto set $\mathcal X^*_t$ of $F_t$ can also be very different from the Pareto set $\mathcal X^*$ of $\sum^T\_{t=1} F_t$. In some cases where $\mathcal X^*_t$ is far from $\mathcal X^*$, the generated decision $x_t$ may even dominate any comparator $x^*_t\in\mathcal X^*$ regarding $F_t$, i.e., $f_t^i(x_t)<f_t^i(x_t^*)$ for any $i\in\\{1,\dots,m\\}$. Since PSG always yields non-negative values, i.e., $\Delta_t\geq0$, your suggested relation $\Delta_t\leq f^i_t(x_t)-f^i_t(x_t^\*)$ does not actually hold for the static regret where $x_t^\*\in\mathcal X^*$.
>
> Consequently, although performing OMD on a single objective $i$ can bound the static regret of this certain objective, namely $R^i(T) = \sum^T\_{t=1} (f^i_t(x_t)-f^i_t(x_t^\*)) \leq O(\sqrt T)$, it may not bound the multi-objective static regret $R\_{\rm MOS}(T)=\sum^T\_{t=1}\Delta_t$, since $\Delta_t\leq f^i_t(x_t)-f^i_t(x_t^\*)$ does not necessarily hold at each round $t$ for the static regret where $x_t^\*\in\mathcal X^*$. In fact, in the Remark below Proposition 1 we show that, when using PSG under $m=1$, $R\_{\rm MOS}(T)$ reduces to $\sum^T\_{t=1} \max\\{F_t(x_t)-F_t(x^*), 0\\}$ where $x^\*\in\text{argmin}\_{x\in\mathcal X}F_t(x)$, which is obviously larger than the standard static regret $R_S(T)=\sum^T\_{t=1} (F_t(x_t)-F_t(x^*))$. Therefore, while OMD optimizes $R_S(T)$, it does not necessarily optimize $R\_{\rm MOS}(T)$ in our setting.
>
> **Q2: “provide a lower bound here … Notably is m^{1/3} tight?”**
>
> **A2**: We have addressed this concern in the **General Author Response (Part 1)**. Notably, we slightly modify our proof and derive a new bound of $O(V_T^{1/3}T^{2/3})$ for OMMD-II, which is now irrelevant of $m$ and thus tighter than our previous bound $O(m^{1/3}V_T^{1/3}T^{2/3})$. We further show that our newly derived bound matches the best attainable bound w.r.t. $V_T$ and $T$ in the single-objective setting, and that our bound is tight w.r.t. $m$.
>
> **Q3: As far as I know, this is a novel result. However, I am not entirely convinced that this is significant enough for the ICLR conference since this Pareto regret bound only holds in the dynamic setting.**
>
> **A3:** Admittedly, we can define the static regret in the multi-objective setting using a new discrepancy metric. In the revision (Section D in the appendix), we provide a possible form of multi-objective static regret, enlightened by the equivalent form of the dynamic regret in Proposition 1. We further prove that it can indeed be optimized by our proposed OMMD-II algorithm. However, the discrepancy metric we use in the regret form is no longer PSG. In fact, it is a bit strange and very complicated to be understood since it can be negative, while most existing Pareto suboptimality metrics always yield non-negative values. Please refer to the **General Author Response (Part 2)** for more detailed discussions.
>
> As the first systematic study of multi-objective online convex optimization, we initially focus on analyzing the dynamic regret, which can be formulated based on the existing zero-order metric PSG and is much easier to be understood. Note that the derivation of the first-order equivalent form of PSG given in Proposition 1 makes our analysis possible, and its significance and non-triviality have also been well approved by Reviewer ieZJ. Moreover, our analysis can also enlighten future research on multi-objective online learning, for example designing the static regret.

---

> > ### Comment · Reviewer_9N6u · 2021-11-30
> > **Re: Author Response**
> >
> > Thanks for your response.
> >
> > I think that the detailed analysis for the static regret, given in Section D, can improve the quality of the paper. Indeed, the fact that this analysis is no longer PSG is indeed quite intriguing, but in essence, we get a bound in $O(\sqrt{T})$, which is reassuring. For the dynamic regret, the fact that it does not depend on $m$ is also an improvement.
> >
> > So, I have modified my review accordingly.

---

### Official Review · Reviewer_ieZJ · 2021-11-03

**Correctness:** 3
**Technical Novelty And Significance:** 4
**Empirical Novelty And Significance:** 4
**Recommendation:** 6
**Confidence:** 4

**Main Review:**

Writing:

The writing of the paper is clear in general and easy to follow.

Novelty:

1. This paper introduces multi-objective online convex optimization, which is a novel and very well-motivated.

2. The performance metric proposed in the paper, i.e., Pareto suboptimality gap (PSG), has been used in the bandit setting. However, in the bandit case, people usually consider directly minimizing PSG through UCB+arm elimination techniques, which can only be applied to the setting when the arm set is **finite**. Thus, it was unclear to me how to generalize the results to the full-information setting. In this paper, the authors do a great job in Proposition 1, which shows an interesting connection between PSG and the dynamic regret and transforms the metric to a max-min form by introducing a weighting vector. Based on this formulation, the authors propose an algorithm which alternatively minimize $\lambda$ and $x_t$ based on regularized MGDA and OMD. I think the formulation in Proposition 1 and the proposed methods are novel.

3. The authors successfully prove the proposed methods enjoy sublinear multi-objective regret, and the experiments also the effectiveness of the proposed methods.

Some questions and minor points:
1. I am a bit confused when reading the proof of Lemma 1: To me, it seems that there should be two sequences of $\lambda_t$: the first sequence of $\lambda_t$ is from Proposition 1, and the second sequence is the $\lambda$ generated by the algorithm. I think the proof of Lemma 1 is mainly related to the second sequence, and my questions is how this related to the first sequence (and also the regret)?

2. It would be great if lower bound wrt m could be obtained.

3. Since $\lambda$ is a m-D metrix and $\nabla F(x_t)$ is a $n\times m$ vector, it would be better to write $\nabla F(x_t) \lambda$ instead of $\lambda\nabla F(x_t)$ in the proof.

4. Eq.(1): $x$ -> $x_t$



**Summary Of The Paper:**

1. This paper introduces the problem of multi-objective online convex optimization.
2. The authors propose to use PSG as the performance metric, and show that it is related to the dynamic regret.
3. The authors develop algorithms that enjoy sublinear multi-objective regret, and also conduct experiments to show the effectiveness of the proposed methods.

**Summary Of The Review:**

I think the proposed problem, regret formulation and methods are interesting and novel, so I vote for acceptance. I have a minor question about the proof, and I hope the authors could help me understand it.

---

> ### Author Response · Authors · 2021-11-23
> **Author Response**
>
> Thank you for your positive and detailed review!
>
> **Q1: “the proof of Lemma 1 is mainly related to the second sequence … how is this related to the first sequence (and also the regret)”**
>
> **A1:** The first sequence of $\lambda_t$ in Proposition 1 is used in the infimum operation over $\lambda_t\in\Delta_m,t\in\\{1,\dots,T\\}$ in the equivalent form of the dynamic regret, thus it is general. The second sequence of $\lambda_t$ in Lemma 1 is specifically generated by OMMD-I or OMMD-II, just as you understand. In the proof of Lemma 1, we utilize the property of the infimum operation, i.e., the regret can be bounded by using an arbitrary sequence of $\lambda_t\in\Delta_m$. Hence to derive the regret bound of the proposed algorithms, we just need to specify the general sequence of $\lambda_t$ to be the specific sequence generated by OMMD-I or OMMD-II.
>
> The reuse of the notations “$\lambda_t$” in both Proposition 1 and Lemma 1 incurs your confusion. In the revision, we have replaced the notations “$\lambda_t$” in Proposition 1 by “$\lambda^*_t$”, indicating that the first sequence is a general sequence.
>
> **Q2: “It would be great if lower bound w.r.t. $m$ could be obtained.”**
>
> **A2:** We have addressed this concern in the **General Author Response (Part 1)**. Notably, we sightly modify our proof and derive a bound of $O(T^{2/3}V_T^{1/3})$ for OMMD-II, which is now irrelevant of $m$ and thus tighter than our previous bound $O(m^{1/3}T^{2/3}V_T^{1/3})$.  Moreover, we further prove the tightness of our newly derived bound w.r.t. $m$ and also illustrate why the lower bound is independent of $m$.
>
> **Q3: “It would be better to write $\nabla F(x_t)\lambda$ instead of $\lambda \nabla F(x_t) $.”**
>
> **A3:** Good suggestion! We have replaced all "$\lambda\nabla F(x_t) $" by "$\nabla F(x_t)\lambda$" in the revision.
>
> **Q4: The typo in Eq (1).**
>
> **A4:** We have revised it in the revision.

---

### Public Comment · ~Alex_Lee2 · 2021-11-15
**It seems that there are some inconsistencies between the statements in the paper and the implementations in the code.**

It seems that there are some inconsistencies between the statements in the paper and the implementations in the code.
For example,
1) In the simulation experiments, the paper said "The learning rates $\eta$ ...... are set as what the corresponding theories suggest", but I found the learning rates $\eta$ are decided via a grid search in the code.
2) In the real-world experiments, the paper said "The learning rates $\eta$ are decided via a grid search ", but I found the they are pre-defined by some magic numbers in the code.

---

> ### Author Response · Authors · 2021-11-17
> **Feedback**
>
> Thank you for your great interest in our paper and code.
>
> In simulation experiments, we did set the learning rates suggested by our theoretical results. The grid search was another approach commonly used to decide hyperparameters in online learning experiments [1]. We have also tried this approach to make comparisons. We left these testing traces in our code unintentionally. Sorry for the confusion.
>
> As reported in our paper, we have used grid search in real-world experiments. The corresponding code was already included in lines 79-85 of adareg_online.py. You seem to have ignored it. Please re-check.
>
> The code in our supplementary materials is an initial version rushed before the ICLR submission deadline. We have not been able to delete unrelated lines and add more comments in time due to the limited time budget. We will re-arrange it carefully in the revision. Currently, please follow the claims in the paper.
>
> If you have any additional questions later, we are very delighted to answer them. You can also look at some other papers in online learning for more reference.
>
> [1] Delay-Tolerant Algorithms for Asynchronous Distributed Online Learning. McMahan and Streeter. NeurIPS 2014.

---

### Author Response · Authors · 2021-11-21
**General Author Response (Part 1)**

We thank all the reviewers for the insightful and detailed reviews. In this post, we would like to address the first major common concern regarding the tightness of the derived bound.

**Common Concern: The tightness of the derived regret bound.**

**Response:** After a careful check, we have found that our previously derived regret bound $O(m^{1/3}T^{2/3}V_T^{1/3})$ was a bit loose w.r.t. $m$. The looseness stems from our derivation of Lemma 1. Specifically, on page 17 of the paper's initial version, we bound $\vert\lambda_t^\top(F_t(x)-F\_{t+1}(x))\vert$ using the following inequality

$\vert\lambda_t^\top(F_t(x)-F\_{t+1}(x))\vert \leq \sup\_{\lambda\in\Delta_m} \vert\lambda^\top(F_t(x)-F\_{t+1}(x))\vert=\max_{1\leq i\leq m}\vert f^i_t(x)-f^i\_{t+1}(x)\vert \leq \sum^m_{i=1} \vert f^i_t(x)-f^i\_{t+1}(x)\vert.$

However, the first inequality fails to utilize the property of the composition weights $\lambda_t$ generated by our algorithm, and the resulting supremum over $\lambda\in\Delta_m$ is thus too loose. Consequently, the summation from $1$ to $m$ in the right part of the last inequality introduces an extra $m$, which finally appears in the derived regret bound.

In the revision, we introduce a novel and non-trivial lemma (Lemma 5) to bound $\vert\lambda_t^\top(F_t(x)-F\_{t+1}(x))\vert$ more tightly, which removes the extra $m$. This treatment tactfully utilizes the stability of the sequence of $\lambda_t$ induced by OMMD-II, illustrating the advantage of our novel algorithm design regarding stabilizing $\lambda_t$. As a consequence, we show that with proper choices of $\eta$ and $\alpha$, OMMD-II attains the regret bound $O(T^{2/3}V_T^{1/3})$, which is now irrelevant of $m$ and thus tighter than our previous bound $O(m^{1/3}T^{2/3}V_T^{1/3})$.

Our newly derived bound $O(T^{2/3}V_T^{1/3})$ matches the best attainable dynamic regret bound $O(T^{2/3}V_T^{1/3})$ w.r.t. $T$ and $V_T$ in the single-objective setting (see references [1, 2, 3] for more detailed discussions). **Moreover, the new bound is now **TIGHT** w.r.t. $\boldsymbol{m}$.**

We corroborate the tightness of $m$ in the following. Consider the scenario where $f^1_t=f^2_t=\dots=f^m_t$ at each round $t$. Then the Pareto set $\mathcal X^*_t $ of the instantaneous loss $F_t$ is identical with the optimal decision set of the scalar loss $f^1_t$, i.e., $\mathcal X^*_t={\rm argmin}\_{x\in\mathcal X}f^1_t(x)$. Now it is easy to verify that PSG becomes $\Delta_t = f^1_t(x_t) - f^1_t(x^*_t)$, where $x^*_t\in\mathcal X^*_t$.  Consequently, we have that the multi-objective dynamic regret $R\_{\text{MOD}}(T) = \sum_t\Delta_t = \sum_tf^1_t(x_t) - f^1_t(x^*_t)$, where $x^*_t \in {\rm argmin}\_{x\in\mathcal X} f^1_t(x)$, which exactly reduces to the dynamic regret regarding the first objective $f^1_t$ in the single-objective setting, which we denote as $R_D(T)$. Hence the lower bound of $R_D(T)$ is also a lower bound of $R\_{\text{MOD}}(T)$ for any $m$. Note that the lower bound of the single-objective dynamic regret $R_D(T)$ is certainly irrelevant of $m$. Thus our newly derived regret bound is tight w.r.t. $m$.

Some readers may suspect it unreasonable that in the multi-objective setting, the derived regret bound does not increase as $m$ increases. Now we explicate the rationality of such independence in the following.

In essence, the independence of m lies in the adoption of PSG in the formulation of the regret. Recall that, in the definition of PSG in Section 2.2, "$\exists i\in\\{1,\dots,m\\}$" means that it just needs to pick one coordinate $i$ to satisfy $f^i_t(x_t)-\epsilon<f^i_t(x'')$, which omits the dependency of $m$. We can see this point from another perspective. Recall that in Proposition 1, PSG at each round $t$ has an equivalent form, namely $\sup\_{x^*_t}\inf\_{\lambda^*_t\in\Delta_m}(\lambda^*_t)^\top(F_t(x_t)-F_t(x^*_t))$, or equivalently $\sup\_{x^*_t}\min\_{i\in\\{1,\dots,m\\}}(f^i_t(x_t)-f^i_t(x^*_t))$. In particular, PSG takes a minimum operation over all objectives, and thus it does not necessarily increase as $m$ increases.

There is another intuitive way that can help understand the rationality of the independence of $m$. As is well recognized in existing research in multi-objective optimization [4], the proportion of the Pareto optimal solutions (or more precisely, non-dominated solutions) in the decision domain tends to increase rapidly as the number of objectives increases. As a consequence, it might not be harder to reach the Pareto optimal set when $m$ turns larger, hence intuitively, the regret bound does not necessarily increase as $m$ increases.


[1] Non-stationary Stochastic Optimization. Besbes et al. Operations Research 2015.

[2] Dynamic Regret of Strongly Adaptive Methods. Zhang et al. ICML 2018.

[3] Optimal Dynamic Regret in Exp-Concave Online Learning. Baby and Wang. COLT 2021.

[4] A Tutorial on Multi-Objective Optimization: Fundamentals and Evolutionary Methods. Emmerich and Deutz. Natural Computing 2018.

---

> ### Author Response · Authors · 2021-11-22
> **General Author Response (Part 2)**
>
> In this post, we would like to address the second major common concern regarding the static regret.
>
> **Common Concern:  Why does our analysis only focus on the dynamic regret without providing a bound for the static regret? How to derive the multi-objective static regret and its corresponding bound?**
>
> **Response:** We divide our response into three main parts. **(1)** We first explain the intrinsic hardness of formulating multi-objective static regret in more detail, indicating the reason why we only focus on the dynamic regret initially. **(2)** Then, by analogy to the proposed multi-objective dynamic regret, we define one version of multi-objective static regret and derive a tight bound w.r.t. it for the proposed algorithm OMMD-II. **(3)** Lastly, we argue that defining such a static regret requires inducing a new discrepancy metric which is a strange generalization of traditional PSG and leaves lots of unfinished work to be discussed further.
>
> As explained in Section 3, especially in Proposition 1 and the Remark below it, the success of our analysis regarding the dynamic regret is due to the establishment of a novel connection between the dynamic regret and the commonly used Pareto discrepancy metric PSG. However, such a connection does not exist for the static regret. If we want to analyze the static regret, we need to devise an essentially new discrepancy metric that can yield negative values. Nevertheless, most of the commonly used discrepancy metrics in multi-objective optimization can not yield negative values. Thus when preparing this paper, we face a design choice: analyze the dynamic regret or analyze the static regret. Analyzing the dynamic regret is quite ready given our established connection but will leave the static one unexplored. Analyzing the static regret is the first thing that comes to most readers' minds but will require designing a novel yet even somewhat strange discrepancy metric that is very complicated to be understood. As the first research on multi-objective online convex optimization, we choose the former, which utilizes the PSG metric that has been widely recognized in related fields.
>
> In the following, we present our attempt to analyze the static regret in the multi-objective setting. We derive one desired static regret analogously to the equivalent form of the dynamic regret derived in Proposition 1. Recall that in the equivalent dynamic regret, the comparator $x^*_t$ is selected from the Pareto set $\mathcal X^*_t$ of the instantaneous loss $F_t$ at each round $t$; in addition, the composition weights $\lambda^*_t$ are determined at each round consecutively. To formulate the static regret, for given $T$, we use a fixed comparator $x^*$ from the Pareto set $\mathcal X^*$ of the cumulative loss $\sum^T\_{t=1}F_t$ and fixed weights $\lambda^*$ at all rounds. Then the resulting regret takes the following form
>
> $R\_{\text{\rm MOS}}(T) = \sup\_{x^\*\in\mathcal X^*} \inf\_{\lambda^\*\in\Delta_m} \sum^T\_{t=1} (\lambda^*)^\top (F_t(x_t)-F_t(x^*))$.
>
> This new regret reduces to the conventional static regret in the single-objective setting. In theory (Theorem 4), we prove that with proper choices of $\eta$ and $\alpha$, our proposed OMMD-II algorithm attains a regret bound of $O(\sqrt T)$, which exactly matches the lower bound of the static regret in the single-objective setting.
>
> Note that the newly introduced static regret is no longer based on PSG. It can be understood as induced from a new discrepancy metric $\delta(x_t;x^*,F_t,\lambda^*) = (\lambda^*)^\top(F_t(x_t)-F_t(x^*))$, where $\lambda^* \in \Delta_m$. We thus have $R\_{\text{\rm MOS}}(T) = \sup\_{x^\*\in\mathcal X^*} \inf\_{\lambda^\*\in\Delta_m} \sum^T\_{t=1}\delta(x_t;x^*,F_t,\lambda^*)$. Such a metric compares the generated decision $x_t$ at each round with the fixed comparator $x^\*\in\mathcal X^\*$.  Hence, in particular, it is able to produce a negative value when $x_t$ dominates $x^*$ regarding the instantaneous loss $F_t$, making it a general extension of PSG. Although this property is desired in the definition of static regret, it is rarely utilized in the discrepancy metrics in multi-objective optimization. Thus it looks a bit strange as a Pareto suboptimality metric and hence its physical meaning needs to be justified further. Besides, there are undoubtedly many other possible ways to define such metrics. Therefore, regarding the multi-objective static regret, much work remains to be done in the future.  We hope our initial attempt paves the way for future research.

---

### Author Response · Authors · 2021-11-23
**Author Response and Paper Revision Done**

We have addressed the reviewers' concerns in the following posts and the paper revision. Our revisions in the paper are all marked with blue. Note that we have only slightly modified the main body of the paper. Most supplementary contents are presented in the appendix.

We are sorry for the delay in the feedback. To answer the insightful questions raised by reviewers, we have been entirely devoted to preparing the material in the past two weeks. We sincerely hope our response addresses the concerns. We are very glad to address any additional questions that may arise. Thank you for your great efforts!

---

### Author Response · Authors · 2021-11-29
**Eagerly look forward to knowing your update**

Respected Reviewers,

The rolling discussion phase is drawing to a close. We eagerly look forward to knowing your update after the initial author response.

We are wondering whether your concerns have been well addressed. If you have any additional questions, it would be great if you could let us know. We are readily prepared to address them.

---

### Decision · Program_Chairs · 2022-01-20

**Decision:**

Reject

**Comment:**

This paper looks at a formulation of online multi-objective optimization problem.

All reviewers agree on the score, 6, which is quite rare but is not really informative; none of them are very excited about the paper, but they all find it interesting.

I have read it as well myself. The paper is rather clear and well written. I have three majors concerns.
1) I am not fully convinced by the objective R_{MOD} as it reduces to the dynamic regret in the single objective problem, as the later cannot be minimized unless we make strong stationarity assumption. This is obviously the case here (see Assumption 2). Then the choice of parameters would depend on some "stationarity" quantity (V_T). I am not really enthusiastic about this either.
2) The analysis is rather classical once the problem is reduced to a single-objective, so the analysis is not really breathtaking. Yet I admit that I quite enjoyed reading about this reduction, the idea is quite neat.
3) Multi-objective online optimization has already been considered in online learning, but the related works did not really mention it. For instance, Blackwell approachability is such an example [1,2,3] (yet I am not sure that it can cover the Pareto front idea). It would be interesting to see how those approach compares (notably, the online mirror descent has been widely studied in that case).

All in all, I do understand the reviewers, and this paper is certainly borderline, but I do not think it reaches the acceptance bar yet. As a consequence, I would rather recommend rejection this year.

[1] J. Abernethy, P. Bartlett, D. Hazan. Proceedings of the 24th Annual Conference on Learning Theory, PMLR 19:27-46, 2011.
[2] V. Perchet. Approachability, regret and calibration: Implications and equivalences, Journal of Dynamics & Games,181-254, 2014.
[3] A. Rakhlin, K. Sridharan, and A. Tewari. Online learning: Beyond regret. Proceedings of the 24th Annual Conference on Learning Theory, PMLR, 19:559–594, 2011.